

# Hydrological modeling using the SWAT Model in urban and peri-urban environments: The case of Kifissos experimental sub-basin (Athens, Greece)

5   Evgenia Koltsida[1], Nikos Mamassis[2], Andreas Kallioras[1]

[1]Laboratory of Engineering Geology and Hydrogeology, School of Mining and Metallurgical Engineering, National Technical University of Athens, Heroon Polytechniou Str. 9, 15780 Zografou, Athens, Greece
[2]Laboratory of Hydrology and Water Resources Development, School of Civil Engineering, National Technical University of Athens, Heroon Polytechneiou Str. 9, 157 80 Zographou, Greece

10   *Correspondence to*: Evgenia Koltsida (evita.koltsida@gmail.com)

**Abstract.** SWAT (Soil and Water Assessment Tool) is a continuous time, semi-distributed river basin model that has been widely used to evaluate the effects of alternative management decisions on water resources. This study, demonstrates the application of SWAT model for streamflow simulation in an experimental basin with daily and hourly rainfall observations to investigate the influence of rainfall resolution on model performance. The model was calibrated for 2018 and validated for 15   2019 using the SUFI-2 algorithm in the SWAT-CUP program. Daily surface runoff was estimated using the Curve Number method and hourly surface runoff was estimated using the Green and Ampt Mein Larson method. A sensitivity analysis conducted in this study showed that the parameters related to groundwater flow were more sensitive for daily time intervals and channel routing parameters were more influential for hourly time intervals. Model performance statistics and graphical techniques indicated that the daily model performed better than the sub-daily model.  The Curve Number method produced 20   higher discharge peaks than the Green and Ampt Mein Larson method and estimated better the observed values. Overall, the general agreement between observations and simulations in both models suggests that the SWAT model appears to be a reliable tool to predict discharge over long periods of time.

## 1 Introduction

Water resource problems, including the effects of urban development, alternative management decisions and future climate 25   oscillation on streamflow and water quality, require a deep understanding and accurate modeling of earth surface processes at the catchment scale in order to be addressed  (Gassman et al., 2014). Experimental catchments provide databases of long-term historical hydrological data which are useful in analyzing the mechanisms governing surface runoff as well as for developing and validating watershed, water quality and water resources management models (Goodrich et al., 2020). They are also able to monitor the major components of the surface hydrological cycle by using remote sensing and geophysical 30   measurements (Tauro et al., 2018). Furthermore, they can monitor groundwater and river water quality with the use of tracer



experiments which can estimate the residence and travel times of water in different components of the hydrological cycle (Hrachowitz et al., 2016; Stockinger et al., 2016).

Bogena et al. 2018 presented an extensive overview of hydrological observatories that are presently operated worldwide with various environmental conditions. The US Department of Agriculture-Agricultural Research Service's (ARS) Experimental

Watershed Network has operated over 600 watersheds in its history and currently operates more than 120 experimental hydrological watersheds (Goodrich et al., 2020). The USDA-ARS watersheds provide deep knowledge of watershed processes and contribute in the development and validation of numerous watershed models.  In addition, many of the watersheds have been used as validation sites for satellite sensors. The Hinkson Creek Watershed (HCW) is an urbanizing agricultural experimental watershed, located in central Missouri, USA. The HCW contributes to the understanding of

precipitation/discharge relationship in multiple-land-use watersheds and investigates the impact of land use on the hydrology regime and nutrient yields (Hubbart et al., 2019; Kellner and Hubbart, 2017; Nichols et al., 2016; Zeiger and Hubbart, 2016). Other well-monitored experimental catchments are the Critical Zone Observatories (CZO) in the Unites States (White et al., 2015), the Terrestrial Environmental Observatories (TERENO) in Germany (Zacharias et al., 2011), the Heihe Watershed Allied Telemetry Experimental Research (HiWATER) in China (Li et al., 2013) and the European Network of Hydrological

Observatories (ENOHA) which is a network of hydrological observatories within Europe (Bogena et al., 2018).

Hydrological and water quality models have been widely used to assess water resource problems and to investigate hydrological processes, land use and climate change impacts and best management practices (Daggupati et al., 2015). In recent decades, various models have been developed to operate in several temporal and spatial scales and with different levels of input data and model structure complexity (Arnold et al., 2015). The SWAT (Soil and Water Assessment Tool)

program is a physically based, semi-distributed, continuous time river basin model (Arnold et al., 2012). The model is an open source code and has five main official versions, SWAT2000, SWAT2005, SWAT2009, SWAT2012, and SWAT+. It has been applied to catchments of various sizes and to several temporal scales (e.g., monthly, daily and sub-daily time step).

SWAT has two methods for the estimation of surface runoff; the SCS Curve Number (CN) method (Soil Conservation Service, 1972) for daily rainfall and the Green and Ampt Mein Larson infiltration (GAML) method (Mein and Larson, 1973)

for sub-daily rainfall. The CN method has been used more often than the GAML method, in SWAT model applications, mainly due to the absence of high temporal resolution data needed for the sub-daily module (Bauwe et al., 2016; Brighenti et al., 2019; Gassman et al., 2014). The few available studies suggest that the calibrated streamflow results are more accurate using the CN approach when compared to the GAML approach (Bauwe et al., 2016; Cheng et al., 2016; Ficklin and Zhang, 2013; Kannan et al., 2007). In particular, in the study where CN improved the results, Kannan et al. (2007) identified a

suitable combination of evapotranspiration and runoff generation methods and reported that the CN method performed better than the GAML method. In contrast, three studies reported that the GAML method simulated better the peak flows during the flood season  than the CN method (Li and DeLiberty, 2020; Maharjan et al., 2013; Yang et al., 2016). Some studies, have pointed out that both approaches have limitations and that the improvement depends on the part of the hydrograph that is analyzed (e.g., high, medium or low flows) and the time scale (e.g., daily, monthly or annually) (Han et al., 2012; King et al.,





1999). Furthermore, several sub-daily applications have been conducted such as land use and management impacts on flood events (Golmohammadi et al., 2017; Campbell et al., 2018), the use of high temporal resolution data for the improvement of the model (Bauwe et al., 2017; Boithias et al., 2017) and modeling of rainfall-runoff events (Jeong et al., 2010; Yu et al., 2018). The authors generally found that finer temporal resolution time steps do not always improve model performance but depend on the basin scale and the characteristics of the watershed. A detailed description of the model history and

applications can be obtained in Gassman et al. (2007), Douglas-Mankin et al. (2010), Brighenti et al. (2019) and Tan et al. (2020).

  In this study, the latest version of SWAT was used to simulate streamflow in an experimental basin using daily and sub-daily (hourly) rainfall observations in order to estimate the influence of rainfall resolution on model performance . To calibrate the model, water level data were obtained from the river gauge located at the basin outlet. The model calibration

and uncertainty assessment were achieved using the Sequential Uncertainty Fitting program (SUFI-2) in SWAT-CUP software (Abbaspour et al., 2004, 2007). The information of the study area, methodology and data input is presented in Section 2, results and discussions are detailed in Section 3 and conclusion is provided in Section 4.

## 2 Materials and methods

### 2.1 Study area

The study area includes the upper part (NW sub-basin) of the Kifissos River basin, located in Athens Greece (Fig. 1). The Kifissos River basin occupies an area of 380 km$^2$ and its route is approximately 22 km, of which at least 14 km are within an urban area.The elevation ranges from 94 m to 1399 m with plains in the south and hills in the north part of the basin. The mean annual temperature is 16.4 °C and the mean annual rainfall across the basin is 577.2 mm.

  The study area is characterized as an urban/sub-urban area, with residential areas, shrubland and agriculture accounting for

34.1, 15.9 and 12.4 % of its land use coverage, respectively (Fig. 2a). It includes mainly four soil types, Cambisols, Regosols, Leptosols and Luvisols (Fig. 2b). The dominant soil formations are characterized by good soil permeability and high contents of clay and sand.





**Figure 1. Geographical location of the study area.**


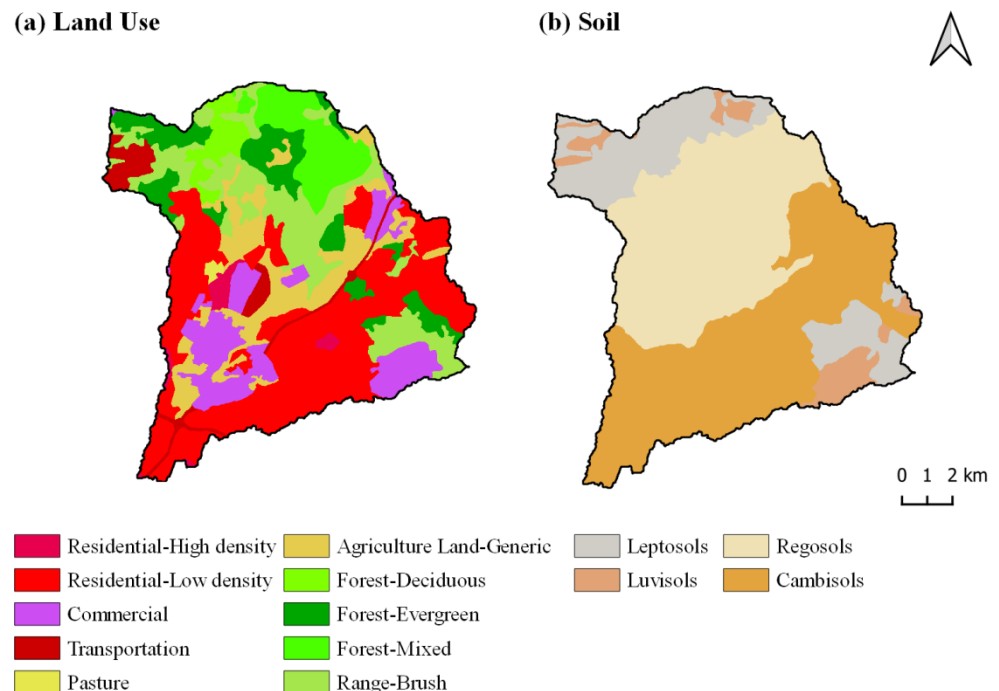

**Figure 2. Spatial distribution of land use and soil.**

## 2.2 Experimental Catchment of Athens Metropolitan Area

The study area includes four in-site monitoring stations measuring water level and water velocity on the river in different times and under different weather conditions (Fig. 1).  The stations were installed at the end of 2017 under the supervision of the School of Mining of NTUA. The network was developed under the European SCENT (Smart Toolbox for Engaging Citizens in a People-Centric Observation Web) program. The station which is located at the outlet of the study area was selected as the most suitable for further analysis in this study, because the three upstream stations were out of order for a

long time and continuous discharge time series were not available for calibration and validation.

The pressure measurement sensor was an Adcon LEV1 Level Sensor (pressure transducer), with 0.1% distance measurement accuracy from the target. The automatic level measurement sensor was a Pulsar dBi intelligent transducer (ultrasonic measurement transducer) with reliable measurement from 125 mm to 15 m. The Stylitis-20 data logger was connected to the sensor. The data logger also offers the possibility to either download the data in situ or remotely transfer data through the use

of the built in GSM/GPRS modem. The time step for transmitting the water level information has been set to 15 minutes. The water level and water velocity data are provided freely from Open Hydrosystem Information Network (OpenHi.net).





### 2.3 Data sources

The input data for the construction of the SWAT model include a digital elevation model (DEM), a land use map, a soil map, and meteorological data (i.e., rainfall, temperature, wind speed, relative humidity and solar radiation). Table 1 summarizes

the input data along with their sources, used in this study.

The digital elevation model (DEM) at 30 m spatial resolution was downloaded from the website of the US Geological Survey (USGS). The land use map was derived from the 100 m 2018 Corine Land Cover map (CLC, 2018) and was modified according to SWAT land use categories (Table 2). The soil map was created from data of the Food and Agriculture Organization (FAO) Digital Soil Map of the World (FAO et al., 2012). In addition, rainfall data, relative humidity, wind

speed, and the minimum and maximum air temperature were obtained from National Observatory of Athens (NOA). Solar radiation data were simulated by WGEN, a weather generator developed by SWAT to fill the missing meteorological data by the use of monthly statistics. A rain gauge network consisting of 5 gauges is distributed throughout the study area as illustrated in Fig. 1. Daily and hourly ($\Delta t$ = 1h) rainfall data were retrieved from 2017 to 2019 with coverage during the entire year. The daily and sub-daily observed streamflow data at the outlet of the basin (Fig. 1) from 2017 to 2019 were

acquired from Open Hydrosystem Information Network (OpenHi.net).

**Table 1. SWAT model input data and sources.**

| Data type | Resolution | Source | Description |
|---|---|---|---|
| DEM | 30 m × 30 m | Shuttle Radar Topography Mission https://earthexplorer.usgs.gov/ | Digital elevation model |
| Land use | 100 m × 100 m | Corine Land Cover https://land.copernicus.eu/ | Land use map |
| Soil | 30 arcseconds (1:5.000.000) | Food and Agriculture Organization, http://www.fao.org/ | Soil map |
| Weather data | 5 gauges | National Observatory of Athens, https://www.meteo.gr/ | Daily data for 2017-2019, sub-daily data for 2017-2019, minimum and maximum air temperatures, relative humidity, wind speed |
| Observed streamflow | 1 gauge | Open Hydrosystem Information Network, https://openhi.net/ | Daily data for 2017-2019, sub-daily data for 2017-2019 |




**Table 2. Land use classification of the Kifissos basin and the corresponding SWAT land use category.**

| CLC code | Corine description | SWAT code | SWAT description | (%) Watershed |
|---|---|---|---|---|
| 121 | Industrial or commercial units | UCOM | Commercial | 11.43 |
| 112 | Discontinuous urban fabric | URLD | Residential-Low Density | 34.11 |
| 122 | Road and rail networks and associated land | UTRN | Transportation | 4.07 |
| 111 | Continuous urban fabric | URHD | Residential-High Density | 1.54 |
| 231 | Pastures | PAST | Pasture | 0.31 |
| 243 | Land principally occupied by agriculture, with significant areas of natural vegetation | AGRL | Agricultural Land-Generic | 12.39 |
| 311 | Broad-leaved forest | FRSD | Forest-Deciduous | 3.11 |
| 312 | Coniferous forest | FRSE | Forest-Evergreen | 9.59 |
| 313 | Mixed forest | FRST | Forest-Mixed | 7.51 |
| 323 | Sclerophyllous vegetation | RNGB | Range-Brush | 15.94 |


## 2.4 Soil Water Assessment Tool (SWAT)

The SWAT (Soil and Water Assessment Tool) program is a semi-distributed, continuous-time, process based model (Arnold et al., 1998, 2012). The model operates on a daily time step and has been developed to evaluate the impact of management practices on water, sediment and agricultural chemical yields in large river basins over long time periods. The main

components of SWAT are hydrology, weather, soil properties, land use, crop growth, sediments, nutrients, pesticides, bacteria and pathogens.

In SWAT, a watershed is divided into multiple sub-basins, which are then subdivided into hydrologic response units (HRUs) based on unique soil, slope and land use attributes. Hydrologic response units (HRUs) enable the model to represent differences in evapotranspiration for various types of vegetation and soil. Simulation of the hydrology of a watershed can be

separated in the land phase, which determines the loadings of water, sediment, nutrients, and pesticides to the main channel, and in the routing phase, which is the movement of the loadings through the streams of the subbasins to the outlets (Neitsch et al., 2011).

Hydrological processes are simulated separately for each HRU, including canopy storage, surface runoff, partitioning of the precipitation, infiltration, redistribution of water within the soil profile, evapotranspiration, lateral subsurface flow from the

soil profile, and return flow from shallow aquifers (Gassman et al., 2007). SWAT uses a single plant growth model to simulate all types of vegetation and is capable to differentiate between annual and perennial plants. The plant growth model estimates the amount of water and nutrients removed from the root zone, transpiration and biomass/yield production.

The main difference between the daily and sub-daily simulation in SWAT occurs in the estimation of surface runoff. The SCS Curve Number (CN) method (Soil Conservation Service, 1972) is used for daily simulations and the Green and Ampt





Mein Larson infiltration (GAML) method (Mein and Larson, 1973) is used for sub-daily simulations. The CN method is an empirical model, widely used, and requires land use, soil, elevation and daily rainfall data as input. The GAML method is a physically based model, uses the same spatial coverages as the CN method, and requires more detailed soil information and sub-daily rainfall records as input. More details on model theory, equations and processes can be found in Arnold et al. (1998), in Gassman et al. (2007) and in Neitsch et al. (2011).

## 2.5 Model setup

The latest version of the SWAT 2012 hydrological model was used in this study. The QSWAT plugin (Dile et al., 2016) embedded in QGIS platform was used for the setup and the parameterization of the model. The watershed delineation, stream parameterization and overlay of soil, land use and slope were automatically completed within the interface. A drainage area of 3.6 km$^2$ was chosen to discretize the study area. The area was delineated into 25 sub-basins, which were

then divided into 175 hydrological response unit (HRUs).

The SWAT models for the Kifissos basin include daily and sub-daily (hourly) rainfall observations. . Potential evapotranspiration was calculated by the Penman-Monteith method, surface runoff was estimated using the CN method for the daily model and the GAML method for the hourly model, and the variable storage coefficient method was used to calculate the channel routing. The simulation period was from 2017 to 2019 and the first year was used as a warm-up period

in order to mitigate the unknown initial conditions. The model was calibrated from 01/01/2018 to 31/12/2018 and validated from 01/01/2019 to 31/12/2019 for discharge, using the SUFI-2 program in SWAT-CUP software (Abbaspour et al., 2004, 2007).

## 2.6 Sensitivity Analysis, Model Calibration and Validation

Watershed models are characterized by large uncertainties related to conceptual design, input data and parameters

(Abbaspour et al., 2015).

The model calibration, validation, and uncertainty analysis were achieved with the use of the SUFI-2 algorithm in the SWAT-CUP software (Abbaspour et al., 2004, 2007). In SUFI-2, uncertainties in parameters (e.g., uncertainty in input data, conceptual model, parameters and measured data) are expressed as ranges or uniform distributions. The concept behind this algorithm is to collect most of the observed data within a narrow uncertainty band. The initial ranges of the calibrating

parameters are set, based on literature and sensitivity analyses. Then, parameter sets are generated using Latin hypercube sampling and the objective function is estimated for each parameter set. The uncertainties are calculated at the 2.5% and 97.5% levels of the cumulative distribution of all output variables, and it is referred to as the 95% prediction uncertainty (95PPU). The goodness of model performance and output uncertainty are assessed using the P-factor and the R-factor (Abbaspour et al., 2004). The P-factor is the percentage of measured data bracketed by the 95PPU band and it ranges from 0

to 1, where 1 means all of the measured data are within model prediction uncertainty. The R-factor is the ratio of the average width of the 95PPU band and the standard deviation of the measured data. The values of R-factor range from 0 to infinity,



where a value near 0 reflects an ideal situation. The spatial scale of the project and the accuracy of the observed data affect the values of the P-factor and the R-factor (Abbaspour et al., 2015). In this study the Nash-Sutcliffe model efficiency (NS) was used as an objective function for both daily and sub-daily calibration and validation. The sensitivities of the parameters

were estimated using the following equation (Eq. 1) (Abbaspour et al., 2015):

$$g = a + \sum_{i=1}^{m} \beta_i b_i, \tag{1}$$

where $g$ is the goal function and $b's$ are the parameters selected for calibration. The sensitivities are calculated as average

changes in the objective function which result from changes in each parameter, while all other parameters are changing. A t-test is then conducted to evaluate the significance of each parameter $b_i$. Parameters with large t-stat and small P-value were characterized as sensitive parameters.

Model validation was achieved using the calibrated parameter ranges without any further changes and the model performance of the calibration period was compared to the model performance of the validation period. The year 2017 was

set as a warm-up period, the streamflow data from the year 2018 were used for calibration and the streamflow data from the year 2019 were used for validation. The statistics on annual precipitation and daily discharge were calculated for each period to overcome biases in discharge patterns. Annual precipitation for 2018 was 566 mm and annual precipitation for 2019 was 735 mm. Mean and standard deviation for 2018 were 1.25 and 0.46 and for 2019 were 1.42 and 0.74 respectively. These statistics ensure that the selected periods represent both wet and dry conditions. In the calibration and validation process, 18

parameters (Table 3) were used. About 600 simulations per iteration were conducted, and up to three iterations, until the results of P-factor and R-factor were satisfying.

Further evaluation of the model performance was achieved with the use of graphical and statistical techniques (Daggupati et al., 2015b; Harmel et al., 2014; Moriasi et al., 2007, 2015). Most commonly used statistical techniques are Nash-Sutcliffe efficiency (NSE) (Nash and Sutcliffe, 1970) coefficient of determination ($R^2$) (Moriasi et al., 2007) and percent bias

(PBIAS) (Gupta et al., 1999) as shown in Eqs. (2), (3), and (4). Most commonly graphical techniques are time series charts, scatter plots, bar charts, maps and percent exceedance probability curves. The statistics were calculated for both models and then their performance was discussed according to guidelines given by (Moriasi et al., 2007, 2015).

$$R^2 = \frac{\left[\sum_{i=1}^{n}(Q_{obs}(i)-\overline{Q}_{obs})(Q_{sim}(i)-\overline{Q}_{sim})\right]^2}{\sum_{i=1}^{n}(Q_{obs}(i)-\overline{Q}_{obs})^2 \sum_{i=1}^{n}(Q_{sim}(i)-\overline{Q}_{sim})^2}, \tag{2}$$

$$NS = 1 - \left[\frac{\sum_{i=1}^{n}(Q_{obs}(i)-Q_{sim}(i))^2}{\sum_{i=1}^{n}(Q_{obs}(i)-\overline{Q}_{obs})^2}\right], \tag{3}$$

$$PBIAS = \left[\frac{\sum_{i=1}^{n}(Q_{obs}(i)-Q_{sim}(i))*100}{\sum_{i=1}^{n}Q_{obs}(i)}\right], \tag{4}$$



where $Q_{obs}$ is the observed discharge, $Q_{sim}$ is the simulated discharge on day i, $\overline{Q}_{obs}$ is the mean of observed discharge and $\overline{Q}_{sim}$ is the mean of simulated discharge. $R^2$ is a measure of how well the variance of measured data is replicated by the model. $R^2$ can range from 0 to 1, where 0 means no correlation and 1 indicates perfect correlation and less error variance. NSE is a measure of how well the simulated values match the observed values. NSE can range from -∞ to 1, where values ≤ 0 show that the observed data mean is a more accurate predictor than the simulated values and 1 is a perfect fit between simulated and observed values. PBIAS, measures the average tendency of the simulated values to be larger or smaller than the observed values. The optimum value is 0, positive values show model underestimation and negative values show model overestimation. More information about the strengths, weaknesses, and usage of the commonly used measures is presented in Moriasi et al. (2015). The SWAT-CUP software is designed mainly for daily, monthly or annually time step. In order to calibrate the sub-daily model, the SUFI-2 files required minor modifications.

**Table 3. Daily and sub-daily SWAT calibrated parameters. The method "r" indicates that the parameter value is multiplied by (1 + a given value), the method "v" indicates that the parameter value is going to be replaced and the method "a" indicates that the parameter is to be added by a given value (Abbaspour et al., 2007).**




| | Parameter | File Ext | Method | Description |
|---|---|---|---|---|
| Surface runoff | CN2 | .mgt | r Relative | Curve number |
| | SURLAG | .bsn | v Replace | Surface runoff lag time |
| Groundwater/Baseflow | ALPHA_BF | .gw | v Replace | Baseflow alpha factor |
| | GW_DELAY | .gw | a Absolute | Groundwater delay |
| | RCHRG_DP | .gw | v Replace | Deep aquifer percolation fraction |
| | REVAPMN | .gw | v Replace | Threshold depth of water in the shallow aquifer for ''revap'' to occur |
| | GW_REVAP | .gw | v Replace | Groundwater ''revap'' coefficient |
| | GWQMN | .gw | v Replace | Threshold depth of water in the shallow aquifer required for return flow to occur |
| Lateral flow | LAT_TTIME | .hru | v Replace | Lateral flow travel time |
| | HRU_SLP | .hru | r Relative | Average slope steepness |
| Channel | OV_N | .hru | r Relative | Manning's "n" value for overland flow |
| | SLSUBBSN | .hru | r Relative | Average slope length |
| | CH_N2 | .rte | v Replace | Manning's ''n'' value for the main channel |
| | CH_K2 | .rte | v Replace | Effective hydraulic conductivity in main channel alluvium |
| Soil | ESCO | .bsn | v Replace | Soil evaporation compensation factor |
| | SOL_K | .sol | r Relative | Saturated hydraulic conductivity of the soil layer |
| | SOL_BD | .sol | r Relative | Moist bulk density |
| | SOL_AWC | .sol | r Relative | Available water capacity of the soil layer |

## 3 Results and Discussion

### 3.1 Parameter's sensitivity analysis and calibration

The most sensitive parameters obtained in daily and hourly simulation are presented in Table 4. Sensitive parameters are characterized by large t-Test and small p-Value. The parameters were characterized as significantly sensitive when the p-value was less than 0.03.

In the daily model, the most sensitive parameters were deep aquifer percolation fraction (RCHRG_DP), groundwater delay

time (GW_DELAY), lateral flow travel time (LAT_TTIME), average slope steepness (HRU_SLP) and moist bulk density (SOL_BD). These parameters were connected to groundwater flow, runoff generation and channel routing. In the sub-daily model, the significantly sensitive parameters were average slope steepness (HRU_SLP), Manning's "n" value for the main channel (CH_N2), effective hydraulic conductivity in main channel alluvium (CH_K2) and lateral flow travel time (LAT_TTIME). These were all related to channel routing.



The choice of model operational time step has an impact on the sensitivity of the SWAT parameters (Jeong et al., 2010). The parameters related to groundwater flow and runoff generation (GW_DELAY, RCHRG_DP) were more sensitive for the daily time intervals and the parameters regarding channel routing (HRU_SLP, LAT_TTIME, CH_N2, CH_K2) were more sensitive for the hourly time intervals. According to Boithias et al. (2017), the CH_N2 parameter is more sensitive at the hourly time step rather than the daily time step, because at the daily time step the flow peak is influenced by other processes

decreasing the sensitivity of the CH_N2. Overall, in both daily and sub-daily models, channel routing was a very important factor for the simulation of the SWAT models.

The sub-daily model is characterized by larger GWQMN and GW_REVAP values than the daily model. GWQMN is the threshold depth of water in the shallow aquifer required for return flow to occur and GW_REVAP controls the water movement from the shallow aquifer into the overlying unsaturated soil layers. As these parameters increase, the rate of

evaporation increases up to the rate of potential evapotranspiration, resulting in a corresponding decrease of the baseflow.

The fitted value of CH_N2 in hourly simulation was $0.11(m^{-1/3}s)$ and was larger than $0.08$ $(m^{-1/3}s)$ in the daily simulation. The CH_N2 parameter affects the rate and the velocity of flow (Boithias et al., 2017). Therefore, the larger CH_N2 value was connected to smaller flow velocity. In addition, the value range for CN2 was smaller for the sub-daily model, leading thereby to lower peak flows. Other differences were average slope steepness (HRU_SLP), average slope length (SLSUBBSN), groundwater delay time (GW_DELAY) and Manning's "n" value for overland flow (OV_N). Their values

were all smaller in sub-daily simulation. The differences between the two models lay mostly in the different runoff estimation methods used by the two models.

It is worth noting that the observations, procedures and assumptions made for this study may affect the results of this study. The values of the calibrated parameters and their sensitivities are influenced by the type and quality of input data, the

conceptual model, the choice of the objective function and inaccuracies in measured input data used for calibration and validation (Abbaspour et al., 2015; Arnold et al., 2012; Polanco et al., 2017).

**Table 4. Daily and sub-daily SWAT calibrated parameters and their sensitivities.**

| Parameters | Initial ranges | | Daily model | | | | Sub-Daily model | | | |
|---|---|---|---|---|---|---|---|---|---|---|
| | | | t-Test | p-Value | Calibrated ranges | | t-Test | p-Value | Calibrated ranges | |
| | Min | Max | | | Min | Max | | | Min | Max |
| CN2 | -0.10 | 0.10 | 0.38 | 0.70 | -0.04 | 0.10 | -0.09 | 0.93 | 0.00 | 0.10 |
| SURLAG | 0.00 | 10.00 | 0.40 | 0.69 | 0.00 | 10.00 | -0.36 | 0.72 | 4.00 | 9.00 |
| ALPHA_BF | 0.00 | 1.00 | -0.15 | 0.88 | 0.05 | 0.69 | -0.23 | 0.82 | 0.50 | 1.00 |
| GW_DELAY | -30.00 | 90.00 | 4.78 | 0.00 | 10.00 | 95.00 | 0.51 | 0.61 | 10.00 | 80.00 |
| RCHRG_DP | 0.00 | 0.50 | 3.44 | 0.00 | 0.00 | 0.50 | 0.14 | 0.89 | 0.11 | 0.40 |
| REVAPMN | 1000.00 | 2000.00 | 1.51 | 0.13 | 990.00 | 1800.00 | 0.49 | 0.62 | 800.00 | 1800.00 |
| GW_REVAP | 0.02 | 0.20 | -1.37 | 0.17 | 0.02 | 0.20 | -0.16 | 0.87 | 0.06 | 0.21 |
| GWQMN | 0.00 | 500.00 | 0.69 | 0.49 | 100.00 | 500.00 | 0.38 | 0.71 | 150.00 | 500.00 |





| | | | | | | | | | | |
|---|---|---|---|---|---|---|---|---|---|---|
| LAT_TTIME | 0.00 | 180.00 | 15.23 | 0.00 | 0.00 | 170.00 | 14.59 | 0.00 | 0.00 | 170.00 |
| HRU_SLP | -0.50 | 3.00 | -3.87 | 0.00 | -0.01 | 3.00 | -3.71 | 0.00 | 0.20 | 2.30 |
| OV_N | -0.50 | 3.00 | -0.94 | 0.35 | -0.30 | 3.00 | -0.73 | 0.47 | -0.05 | 2.00 |
| SLSUBBSN | -0.20 | 0.20 | 2.11 | 0.04 | -0.10 | 0.20 | 0.89 | 0.37 | -0.06 | 0.20 |
| CH_N2 | 0.01 | 0.30 | 0.09 | 0.93 | 0.01 | 0.20 | 6.52 | 0.00 | 0.03 | 0.20 |
| CH_K2 | 0.00 | 127.00 | -0.83 | 0.41 | 0.00 | 80.00 | 3.52 | 0.00 | 0.00 | 50.00 |
| ESCO | 0.50 | 0.95 | -0.43 | 0.67 | 0.50 | 0.95 | -1.35 | 0.18 | 0.50 | 0.95 |
| SOL_K | -0.80 | 0.80 | -0.94 | 0.35 | -0.20 | 0.80 | -1.98 | 0.05 | -0.10 | 0.68 |
| SOL_BD | -0.30 | 0.30 | -5.69 | 0.00 | -0.10 | 0.30 | -1.31 | 0.19 | -0.01 | 0.27 |
| SOL_AWC | -0.05 | 0.05 | -1.53 | 0.13 | -0.03 | 0.03 | -0.90 | 0.37 | -0.03 | 0.02 |

## 3.2 Daily and sub-daily model performances

Quantitative statistics and criteria recommended by Moriasi et al. (2007, 2015) were used to evaluate the model performance. Figure 3 shows the temporal dynamics of the hydrographs reproduced by both infiltration methods. The high flow season is observed during winter and spring. The low flow season is observed in summer and early fall due to high evapotranspiration. Figure 4 presents the flow duration curves of the two models, indicating good agreement between observed and simulated values. Generally, in the sub-daily model, the simulated discharge peaks did not always match the observed values and were sometimes considerably lower.

The performance statistics are illustrated in Table 5 and indicate reasonable calibrated models for both infiltration approaches. Model performance using the CN method showed better results than the GAML method. In particular, the NSE and $R^2$ indices for the daily model were 0.84 and 0.79 for the calibration period and 0.87 and 0.86 for the validation period. For the sub-daily model the NSE and $R^2$ indices were 0.53 and 0.49 for the calibration period and 0.63 and 0.6 for the validation period respectively. Furthermore, the daily model showed smaller modeling uncertainties with P-factor 0.79 and R-factor 1.58 (compared to 0.83 and 1.71 respectively for the sub-daily model).

Overall, the general agreement between the observed and the simulated values during the calibration and the validation period indicate that the choice of the calibration and validation periods was relevant. According to Moriasi et al. (2015) model performance can be evaluated as "satisfactory" for flow simulations if daily, monthly, or annual $R^2 > 0.60$, NSE > 0.50, and PBIAS $\leq \pm15\%$ for watershed-scale models. These ratings should be modified to be more or less strict based on evaluation time step. Typically, model simulations are poorer for shorter time steps than for longer time steps (e.g., daily versus monthly or yearly) (Engel et al., 2007). Considering these guidelines, the daily and sub-daily models showed satisfactory performance for both calibration and validation periods.

The better performance of the CN method in comparison to the GAML method in this study is consistent with the results of other studies (Bauwe et al., 2016; Ficklin and Zhang, 2013; Kannan et al., 2007; King et al., 1999). Bauwe et al. (2016) evaluated both CN and GAML methods and highlighted that the CN method performed slightly better than the GAML



method. Ficklin and Zhang (2013) generally suggested that for daily simulations the CN method predicted more accurately streamflow as compared to the GAML model. Kannan et al. (2007) identified a suitable combination of ET runoff generation methods and reported that the CN method performed better than the GAML method. Kannan et al. (2007) conducted a sensitivity analysis to identify the best combination of evapotranspiration and runoff method for hydrological modeling and concluded that the CN method performed better than the GAML method for streamflow because the GAML method tends to hold more water in the soil profile and predict a lower peak runoff rate. King et al. (1999) concluded that the GAML method appeared to have more limitations in accounting for seasonal variability than the CN method.

In this study, the CN method produced higher discharge peaks than the GAML method and generally estimated better the observed values. The cause of these results could be that the choice of the sub-daily precipitation time step might be too large for this case. The selection of sub-daily precipitation input time step has a great impact on model results when using the GAML method and it should be based on the scale and characteristics of the watershed (Bauwe et al., 2016; Jeong et al., 2010; Kannan et al., 2007).





**Figure 3. Observed and simulated discharge (m$^3$ s$^{-1}$) at the daily time step (a, b) and at the hourly time step (c, d).**





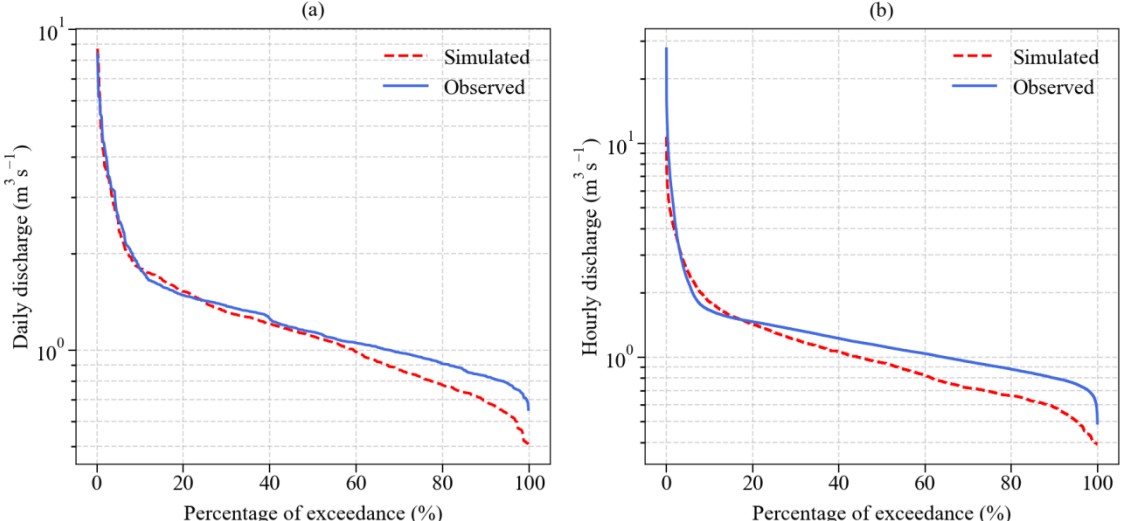

**Figure 4. Observed and simulated flow duration curves (m³ s⁻¹) at the daily time step (a) and at the hourly time step (b).**


**Table 5. Model evaluation statistics of the daily and sub-daily SWAT models for the calibration and validation periods.**

| Time-step | Period | p-Factor | r-Factor | $R^2$ | NS | PBIAS(%) |
|---|---|---|---|---|---|---|
| Daily | Calibration | 0.74 | 1.41 | 0.84 | 0.79 | 6.4 |
| | Validation | 0.79 | 1.58 | 0.87 | 0.86 | 4.2 |
| Sub-Daily | Calibration | 0.72 | 1.33 | 0.53 | 0.49 | 16.9 |
| | Validation | 0.83 | 1.71 | 0.63 | 0.6 | 11.7 |

**3.3 Selected heavy rainfall events**

Heavy rainfall events that occurred in the years 2018 and 2019 (Tatoi station, NOA records) were investigated in order to
examine the accuracy of the sub-daily model and the impact of rainfall on an urban watershed. The hydrographs of the
selected heavy rainfall events are presented in Figure 5.

The first event (Fig. 5a) is the precipitation of 32 mm during January 12 through January 14, 2018. On January 13th, the
observed peak flow reached 10.6 m³/s at 4 am, 10.1 m³/s at 5 am and 10.7 m³/s at 6 am and the simulated peak flow were
5.8, 7.1 and 5.7 m³/s respectively. The average observed discharge rate was 2.6 m³/s and the average simulated discharge
rate was 2.2 m³/s. The second event (Fig. 5b) is the precipitation of 27 mm during May 5 through May 7, 2018. The average
observed discharge rate was 2.2 m³/s and the average simulated discharge rate was 2.1 m³/s. On May 6th, the observed peak
flow was 8.3 m³/s at 19 pm, 11 m³/s at 20 pm and 8.8 m³/s at 21 pm and the simulated peak flow were 6.1 m³/s, 4 m³/s and
6.5 m³/s at the same time. The third event (Fig. 5c) is the precipitation of 56 mm during September 29 through October 1,





2018. About 31.4 mm were recorded from September 29th-10 am to September 30th-0 am. The average observed discharge

rate was 5.7 m³/s and the average simulated discharge was 5.2 m³/s. On September 29th, the observed peak flow reached 14.5

m³/s at 16 pm, 15.8 m³/s at 17 pm and continued to 17.2 m³/s at 18 pm. The simulated peak flows were 7.2, 6.1 and 8.9 m³/s

respectively. On September 30th, the peak flow reached to 10.1, 11.2, 12.1 m³/s at 18, 19 and 20 pm and model simulated

peak flow were 5.5, 7.8 and 9.5 m³/s.

The forth event (Fig. 5d) is the precipitation of 47.6 mm during February 5 through February 7, 2019. The simulated and

observed discharge reached to peak simultaneously but with different magnitude values. Specifically, on February 6th, the

peak flow reached to 7.5, 16.2 and 13.8 m³/s at 0, 1 and 2 am and model simulated peak flow were 6.2, 4.2 and 5.8 m³/s. The

average observed discharge rate was 3.6 m³/s and the average simulated discharge was 2.9 m³/s. The fifth event (Fig. 5e) is

the precipitation of 46.6 mm during November 12 through November 14, 2019. On November 13th, peakflow reached a peak

of 12.3 m³/s at 3 am but the model underestimated peak flow reaching only 3.5 m³/s. On the same day, the peak flow reached

to 9.3, 10.6 and 9.9 m³/s at 9, 10 and 11 am and the model simulated peak flow were 6.2, 7.4 and 6.7 m³/s. The average

observed discharge rate was 2.9 m³/s and the simulated discharge rate was 2.4 m³/s. The sixth event (Fig. 5f) is the intensive

precipitation of 99.6 mm during December 29 through December 31, 2019. On December 30th, the average observed

discharge rate at the outlet gage was 4.9 m³/s, peak flow reached to 13.8 m³/s at 20 pm, continued to 14.8 m³/s at 21 pm and

then discharge started to fall. The average simulated discharge was 3.6 m³/s and the peak flow reached 5.4 and 6.9 m³/s at 20

and 21 pm respectively.

Generally, the hourly model underestimated the peak flows with values much lower than the observations for the majority of

the events. Observational errors in the model input data may explain the difference between the simulated and observed

values as these errors can generate variability, lead to undesired trends, and influence the model calibration and validation

results. Hydrological models are climate-driven, so of the many types of input data, correct representation of spatial

precipitation is essential (Guzman et al., 2015, Kamali et al., 2017).



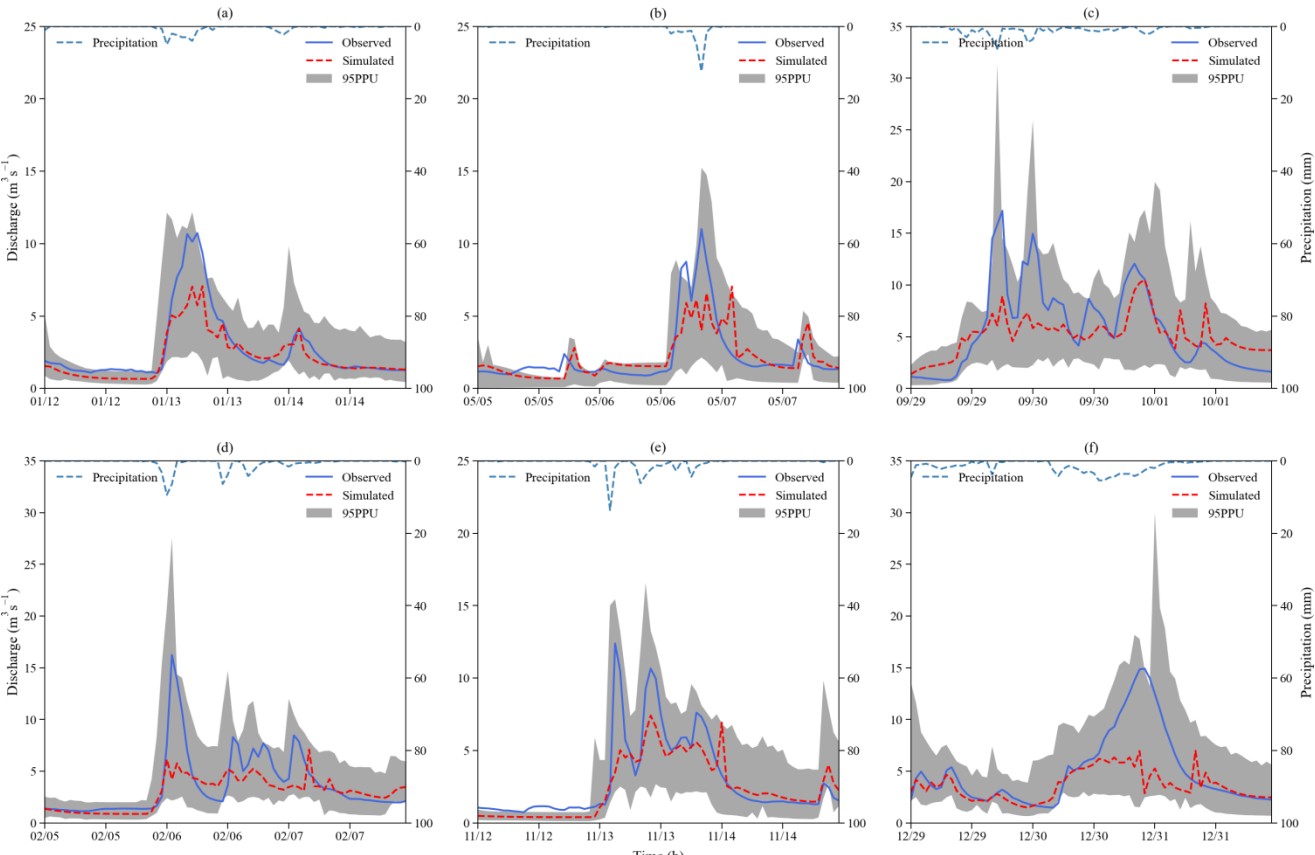

**Figure 5. Observed and simulated hourly discharge (m³ s⁻¹) for the heavy rainfall events that occurred in 2018 and 2019: (a) event from 12/01-14/01/2018; (b) event from 05/05-07/05/2018; (c) event from 29/09-01/10/2018; (d) event from 05/02-07/02/2019; (e) event from 12/11-14/11/2019; (f) event from 29/12-31/12/2019.**

## 4 Conclusions

Experimental catchments provide long term time series of hydrological data which are essential for improved application of best management practices and the development and validation of watershed models. In this study, discharge was monitored for three years (2017-2019) in an experimental basin, located in Athens, Greece. Discharge simulation, calibration and validation were achieved with the application of SWAT model, which has been increasingly used to support decisions on various environmental issues and policy directions. Daily and hourly rainfall observations were used as inputs to SWAT and the model was tested for the period 2017-2019. Surface runoff was estimated using the CN method for the daily model and the GAML method for the hourly model.

A sensitivity analysis conducted in this study showed that the parameters related to groundwater flow were more sensitive for daily time intervals and channel routing parameters were more influential for hourly time intervals. These findings



indicate that the model operational time step affect parameters' sensitivity to the model output, thus demonstrating the need for different model strategy for the simulation of sub-daily hydrological processes.

Generally, the daily model performed better than the sub-daily model in simulating runoff. The CN method produced higher discharge peaks than the GAML method and estimated better the observed values. Quantitative statistics of the observed and

the simulated records indicate that the calibration and validation processes produced acceptable results for both infiltration methods. Additionally, graphical techniques at the outlet station show that both models succeed in capturing majority of seasonality and peak discharge. The differences in the calibrated values of the two models lay mostly in the different runoff estimation methods used by the two models.

Overall, the general agreement between observations and simulations in both models suggests that the SWAT model appears

to be a reliable tool to predict discharge over long periods of time. It should be noted that several factors such as data limitation, observational errors in input data, complexities of spatial and temporal scales and inaccuracies in model structure may lead to uncertainty in model outputs. In the future, emphasis will be placed in the quantification of the parameter uncertainty by including more observed variables in the calibration process such as evapotranspiration and soil moisture satellite data.


*Code availability.* The source codes of the SWAT model are available at the website http://swat.tamu.edu/ (USDA Agricultural Research Service and Texas A&M AgriLife Research)

*Data availability.* The DEM data were downloaded from the website https://earthexplorer.usgs.gov/ (Shuttle Radar Topography Mission,

SRTM). The land use data were downloaded from the website https://land.copernicus.eu/ (Corine Land Cover, CLC 2018). The soil data were downloaded from the website http://www.fao.org/ (Food and Agriculture Organization, FAO). The weather data were downloaded from the website https://www.meteo.gr/ (National Observatory of Athens, NOA). The discharge data were downloaded from the website https://openhi.net/ (Open Hydrosystem Information Network).

*Author contributions.* EK performed the simulations, analyzed the results and prepared the manuscript with contributions from all the co-authors. NM and AK contributed to the conception and methodology of this study. AK was the supervisor of the research project and provided the funding that lead to this publication.

*Competing interests.* The authors declare that they have no conflict of interest.


*Acknowledgements.* The research work was supported by the Hellenic Foundation for Research and Innovation (HFRI) under the HFRI PhD Fellowship grant (Fellowship Number: 1586). The authors are grateful to Nancy Sammons, Information Technology Specialist of the SWAT team and Chris George, Software Engineer of the SWAT team for their help with the SWAT software.



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
