# Peer review of "Hydrological modeling using the SWAT Model in urban and periurban environments: The case of Kifissos experimental sub-basin (Athens, Greece)"

_Hydrology and Earth System Sciences, 2021_

## Author Comment (AC1)

**Reply to comments of Anonymous Referee #1**

We thank anonymous Referee #1 for reviewing our manuscript. The authors are grateful for the insightful comments which provide great suggestions to improve the manuscript.

**General comment**

The paper is interesting and pleasant to read. I have only minor doubts that could be addressed before publication.

Reply: We thank anonymous Referee #1 for the positive feedback and encouragement.

**Specific comments**

Comment: First of all, in the introduction (lines 33-45) the Authors provide an overview of experimental watersheds dealing with hydrological observations. Although the topic is interesting, I suggest to shorten this paragraph because the cited watersheds are not considered in the present manuscript and the topic could be misleading for the reader.

Reply: We agree with the referee and we will merge the first two paragraphs. We will make the following changes in the manuscript:

**Lines 24-45**: "Water resource problems, including the effects of urban development, alternative management decisions, and future climate oscillation on streamflow and water quality, require a deep understanding and accurate modeling of earth surface processes at the catchment scale to be addressed (Gassman et al., 2014). In order to understand catchment processes, it is necessary to obtain detailed weather data and catchment observations related to runoff, water stage, erosion, soil moisture, and water quality. Experimental catchments are properly designed and well-monitored catchments that aim to provide databases of long-term historical hydrological data, which help analyze the mechanisms governing surface runoff (Goodrich et al., 2020). In addition, experimental catchments contribute in the development and validation of numerous watershed models and can be used as validation sites for satellite sensors (Tauro et al., 2018). Furthermore, experimental catchments can monitor groundwater and river water quality with the use of tracer experiments which can estimate the residence and travel times of water in different components of the hydrological cycle  (Hrachowitz et al., 2016; Stockinger et al., 2016). Bogena et al. 2018 presented an extensive overview of hydrological observatories that are presently operated worldwide with various environmental conditions. Among those, the US Department of Agriculture-Agricultural Research Service's (ARS) Experimental Watershed Network has operated over 600 watersheds in its history and currently operates more than 120 experimental hydrological watersheds (Goodrich et al., 2020)".

Comment: Second, in introduction (lines 72-73) I believe that aims and novelty of the manuscript should be more emphasized.

Reply: We agree with the referee that the aims and novelty of this study should be more emphasized. We will make the following changes in the manuscript:

**Lines 72-73**: "In this study, the SWAT 2012 model (rev 681) in the QSWAT interface was used to simulate streamflow in an experimental basin using daily and sub-daily (hourly) rainfall observations. The main objectives were to (i) calibrate and validate the SWAT model using streamflow data, (ii) examine which parameters are more sensitive in different time steps, (iii) estimate the influence of rainfall resolution on model performance, (iv) compare the Curve Number method and Green and Ampt Mein Larson method for runoff simulation, (v) examine the accuracy of the sub-daily model and compare the peak discharges and time of peak of the two models in selected rainfall events, and (vi) investigate the suitability of the SWAT model for hourly simulation in a mixed-land-use basin (i.e., blended combinations of land use). Hence, this study will provide essential hydrological knowledge and contribute to the understanding of the earth surface processes of an urban/peri-urban hydrological system with complex land use in order to analyze the mechanisms governing surface runoff at the catchment scale".

Comment: Third, in paragraph 2.2, I found a little confusing the instruments description and the data that later are used in the manuscript. The Authors mention (lines 101-106) water level and water velocity sensor that are installed in the experimental watershed, but specify only units in mm, what about the velocity? Is the sensor present and used?

Reply: We thank the referee for this comment. This paragraph needs more clarification. The river gauge at the outlet of the basin measures water level data at a 15 min time step. Then, using the Manning's equation we calculated the flow velocity and the flow rate.

Manning's Equation:

$$Q = V * A = \left(\frac{1}{n}\right) * A * R^{\frac{2}{3}} * S^{\frac{1}{2}}$$

Where:

Q = Flow Rate, $(m^3/s)$

V = Velocity, (m/s)

A = Flow Area, $(m^2)$

n = Manning's Roughness Coefficient

R = Hydraulic Radius, (m)

S = Channel Slope, (m/m)

We will exclude the reference to the water velocity in the revised manuscript since we didn't use water velocity in the calibration process. We will also shorten the instruments description.

**Lines 95-106**: "The study area includes four water level monitoring stations that provide continuous recordings of the river stage at pre-selected time-intervals (15mins time-step) (Fig. 1). The stations were installed at the end of 2017 under the supervision of the School of Mining of National Technical University of Athens (NTUA). The network was developed under the EU H2020 RIA Program SCENT (Smart Toolbox for Engaging Citizens in a People-Centric Observation Web). The station which is located at the outlet of the study area was selected as the most suitable for further analysis in this study, because the three upstream stations experienced some mechanical problems that affected the calibration and validation process. The monitoring stations are part Open Hydrosystem Information Network (OpenHi.net) which is a national integrated information infrastructure for the collection, management and free dissemination of hydrological data (OpenHi.net) in Greece."

Comment: Fourth (lines 296-297), the Authors mention an interesting effect of precipitation time step that could affect the result, but in my opinion the results could be affected also by the classic difficulty in obtaining reliable estimations of GAML parameters based on the soil type and heterogeneity. Eventually this issue could be discussed here.

Reply: We thank the referee for this suggestion. The GAML method requires indeed detailed soil data which can be difficult to obtain and may affect the accuracy of the model's results. Hence, we will include the referee's suggestion in the revised manuscript. We will rephrase lines 296-297 in the following way:

**Lines 295-299**: "In this study, the daily model produced higher discharge peaks than the hourly model and generally estimated better the observed values. These results could be due to drawbacks of the GAML method, such as the requirement for detailed soil information and high resolution rainfall data in a sub-daily time step (King et al., 1999). The GAML method assumes that the soil profile is characterized by homogeneity and that the previous soil moisture is distributed uniformly in the soil profile (Jeong et al., 2010). Therefore, the uncertainty in the resolution of the rainfall data, the heterogeneity of the soil formations and the upcoming difficulty in determining the parameters' values for parameterization could affect the method's efficiency. The selection of sub-daily precipitation input time step as well as the resolution of the precipitation data have a great impact on model results when using the GAML method and it

should be based on the scale and characteristics of the watershed (Bauwe et al., 2016; Jeong et al., 2010; Kannan et al., 2007)."

Comment: Fifth (line 337) the Authors mention observational errors; could they specify if this could be attributed to the estimation of channel and hillslope flow velocities?

Reply: We thank the referee for highlighting this point. In the manuscript we mentioned that several factors may lead to uncertainty in model outputs such as observational errors in model input data (i.e., weather, soil and land use data). These errors could be due to inaccuracies in the nature of the sensor, environmental conditions and data collection (Guzman et al., 2015). Other reasons may be data limitation, complexities of spatial and temporal scales and inaccuracies in model structure (Polanco et al., 2017). We didn't discuss though the observational errors associated specifically with the estimation of channel and hillslope flow velocities during the calibration process.

Channel and hillslope velocities define the time of the peak and the shape of the hydrograph. In the SWAT model, the CH_N2 parameter (Manning's "n" value for the main channel) affects the rate and the velocity of the flow (Boithias et al., 2017). In this study, the CH_N2 parameter showed the highest sensitivity in the hourly model. This outcome is similar to those of previous studies (Boithias et al., 2017; Jeong et al., 2010). In particular, according to Boithias et al. (2017) the CH_N2 parameter is more sensitive at the hourly time step rather than the daily time step, because at the daily time step the flow peak is influenced by other processes decreasing the sensitivity of the CH_N2. Therefore, the lower peak flows of the hourly model comparing to the daily model could be attributed to the different calibrated value of the CH_N2 parameter.

The estimation of the channel and hillslope velocities should be included in the potential observational errors among other errors such as the quality of the precipitation, soil and, land use data. We thank the referee and we will include this suggestion in the revised manuscript. We will make the following changes in the manuscript:

**Lines 337-340**: "Furthermore, observational errors in the model input data (i.e., weather, soil and land use data) include inaccuracies in the estimation of channel and hillslope velocities and channel geometry, in the nature of the sensor, environmental conditions and data collection (Guzman et al., 2015). These errors can generate variability, lead to undesired trends, and influence the model calibration and validation results (Kamali et al., 2017). In addition, the complex land use characteristics and processes of an urban/peri-urban environment and assumptions made during the model structure/parameterization process (e.g., selection of parameters for calibration, objective function, and conceptual simplifications) increase the uncertainty of the results."

**Minor comments**

Comment: line 81: route?

Reply: We thank the referee for this comment. We rephrased line 81 in the following way:

**Line 81**: "The Kifissos River basin occupies an area of 380 km$^2$. Kifissos River route is approximately 22 km, of which at least 14 km are within an urban area".

Comment: line 97: specify the acronym.

Reply: We thank the referee for this comment. We added the acronym and changed line 97 in the following way:

**Line 97**: "The stations were installed at the end of 2017 under the supervision of the School of Mining of National Technical University of Athens (NTUA)".

Comment: line 190: changing or constant?

Reply: According to Abbaspour et al. (2017, 2007) the sensitivities are estimated as average changes in the objective function which result from changes in each parameter, while all other parameters are changing.

Comment: line 198: do the numbers refer to discharge?

Reply: We thank the referee for this comment. The numbers refer to discharge. We changed line 198 in the following way:

**Line 198**: "Mean and standard deviation of discharge for 2018 were 1.25 and 0.46 and for 2019 were 1.42 and 0.74 respectively".

References

Abbaspour, K., Vaghefi, S. and Srinivasan, R.: A Guideline for Successful Calibration and Uncertainty Analysis for Soil and Water Assessment: A Review of Papers from the 2016 International SWAT Conference, Water, 10(1), 6, doi:10.3390/w10010006, 2017.

[revised manuscript text omitted]

---

## Author Comment (AC2)

**Reply to comments of Anonymous Referee #2**

We thank anonymous Referee #2 for reviewing our manuscript. The authors are grateful for the insightful comments which provide great suggestions to improve the manuscript.

Comment: Line 26: "Experimental catchments" What this means? It is not clear what you are trying to say. Rephrase the sentences.

Reply: We agree with the referee that the term "Experimental catchments" needs further explanation in the manuscript. We will rephrase the following sentences:

**Lines 24-28**: "Water resource problems, including the effects of urban development, alternative management decisions, and future climate oscillation on streamflow and water quality, require a deep understanding and accurate modeling of earth surface processes at the catchment scale to be addressed (Gassman et al., 2014). In order to understand catchment processes, it is necessary to obtain detailed weather data and catchment observations related to runoff, water stage, erosion, soil moisture, and water quality. Experimental catchments are properly designed and well-monitored catchments that aim to provide databases of long-term historical hydrological data, which help analyze the mechanisms governing surface runoff (Goodrich et al., 2020).

Comment: Line 28: They are? Who are they? There is two times the sentence starting with "they" but it is not clear who or what they are.

Reply: We agree with the referee that the sentence needs clarification. The word "they" refers to the term experimental catchments. We will merge the first two paragraphs and we will make the following changes in the manuscript:

**Lines 24-45**: "Water resource problems, including the effects of urban development, alternative management decisions, and future climate oscillation on streamflow and water quality, require a deep understanding and accurate modeling of earth surface processes at the catchment scale to be addressed (Gassman et al., 2014). In order to understand catchment processes, it is necessary to obtain detailed weather data and catchment observations related to runoff, water stage, erosion, soil moisture, and water quality. Experimental catchments are properly designed and well-monitored catchments that aim to provide databases of long-term historical hydrological data, which help analyze the mechanisms governing surface runoff (Goodrich et al., 2020). In addition, experimental catchments contribute in the development and validation of numerous watershed models and can be used as validation sites for satellite sensors (Tauro et al., 2018). Furthermore, experimental catchments can monitor groundwater and river water quality with the use of tracer experiments which can estimate the residence and travel times of water in different components of the hydrological cycle  (Hrachowitz et al., 2016; Stockinger et al., 2016). Bogena et al. 2018 presented an extensive overview of hydrological observatories that are presently operated worldwide with various environmental conditions. Among those, the US Department of Agriculture-Agricultural Research Service's (ARS) Experimental Watershed Network has operated over 600 watersheds in its history and currently operates more than 120 experimental hydrological watersheds (Goodrich et al., 2020)".

Comment: Line 47-49: Which are these models? Why did you choose SWAT?

Reply: The referee is right to point out this issue. Hydrological models can be categorized as (i) point-scale models, (ii) field-scale models, and (iii) watershed-scale models (Arnold et al., 2015; Moriasi et al., 2007). Point-scale models (i.e., SHAW, COUPMODEL, SWIM3, MACRO, and HYDRUS) are usually used to simulate the physical or chemical processes that occur at a soil profile. Field-scale models (i.e., DRAINMOD, ADAPT, EPIC, DAISY, and, RZWQM2) represent the basic processes of hydrology, soil erosion, vegetation, sediment transport, and pesticides occurring in the combined soil-water-plant system. Watershed-scale models (i.e., SWAT, APEX, HSPF, WAM, KINEROS and, MIKE-SHE) incorporate processes of more spatial and temporal complexity and divide basins into sub-basins, response units or cells.

Among those watershed-scale models, the SWAT model (Arnold et al., 2012) was used in this study. The SWAT model is an open-source software, and it is supported by online documentation and a literature database (Gassman et al., 2014, 2007; Tan et al., 2020). The application of the model involves the division of the hydrological basin into sub-basins and then into Hydrological Response Units (HRU) (Neitsch et al., 2011). In this way, different values of rainfall, temperature, and evapotranspiration, different crops, and different soil types can be simulated. Furthermore, SWAT can be linked with QGIS, which is a free and open-source platform. QSWAT (Dile et al., 2016), which was used in this study, prepares the inputs easily for SWAT, has a friendly user interface, and has the ability to visualize the results, which can be helpful for the interpretation of the many SWAT outputs. The study area is characterized as a mixed-land-use area with high complexity and spatial distribution of input data. The use of a semi-distributed model, such as SWAT, is considered the most appropriate choice because SWAT enables the simulation of as wide a range of processes as possible. Therefore, SWAT was considered a credible tool for discharge simulation for this study area.

We referred to watershed-scale models specifically. We will rewrite the paragraph as follows:

**Lines 46-52**: "Hydrological and water quality models have been widely used to assess water resource problems and to investigate hydrological processes, land use and climate change impacts and best management practices (Daggupati et al., 2015). In recent decades, various watershed-scale models (i.e., SWAT, APEX, HSPF, WAM, KINEROS and, MIKE-SHE) have been developed to operate with different levels of input data and model structure complexity (Arnold et al., 2015; Moriasi et al., 2007). Among the above watershed-scale models, the SWAT program (Soil and Water Assessment Tool) (Arnold et al., 2012) was selected for this study. SWAT is a physically based, semi-distributed, continuous time river basin model and has five main official versions, SWAT2000, SWAT2005, SWAT2009, SWAT2012, and SWAT+. It was selected because is an open source code, has a wide range of online documentation and literature database and has been applied to catchments of various sizes and to several temporal scales (e.g., monthly, daily and sub-daily time step) (Gassman et al., 2014, 2007; Tan et al., 2020). Furthermore, it can be linked to QGIS, an also free and open-source platform, and has the ability to visualize the results, which can be helpful for the interpretation of the many SWAT outputs (Dile et al., 2016)".

Comment: Line 72: Are you used SWAT+?

Reply: We thank the referee for the correction. We used SWAT 2012 (rev 681) in the QSWAT interface. We will rewrite the sentence in the revised manuscript to be as follows:

Line 72: "In this study, the SWAT 2012 model (rev 681) in the QSWAT interface was used to simulate streamflow in an experimental basin using daily and sub-daily (hourly) rainfall observations".

Comment: Line 72: The aim and innovation of the work need to be better discussed both in introduction and discussion. There is plenty of work which evaluates model performances with different data resolution (soil, morphology and climate. Here some examples which can be useful in discuss the main innovation: How you work give new insight in the research? I cannot see any novelty or secondary elaboration from the canonical SWAT application such as a susceptibility map or future prediction. I suggest to the authors to focus more on this points.

https://doi.org/10.5194/hess-24-3603-2020

https://doi.org/10.1016/j.jenvman.2020.110625

https://doi.org/10.5194/hessd-7-4411-2010

Reply: We agree with the referee that the aims and novelty of this study should be more emphasized. This study aims to investigate the complex hydrological processes that take place within a catchment of highly-variable land-use characteristics and the impact of the above on the generation of surface runoff. More specifically the catchment land use/land cover type starts from (i) being almost exclusively rural within its most upstream part, (ii) to peri-urban land use/land cover type within its intermediate part and finally (iii) to urban within its most downstream part. We will re-organize Section 3.2 and Section 3.3 appropriately while in addition to that, we will analyze and discuss the physical meaning of the results in the revised manuscript. We will also provide further description of the objectives and the contribution of this study in the Abstract, Introduction, Results and Discussion sections. In addition, we will explain the peak flow underestimation and we will discuss the disadvantages of the GAML method. We will also aggregate the hourly results to a daily time step in order to compare the performances in the revised manuscript.

We will make the following changes in the Abstract section in the revised manuscript:

Lines 11-22: "SWAT (Soil and Water Assessment Tool) is a continuous time, semi-distributed river basin model that has been widely used to evaluate the effects of alternative management decisions on water resources. This study examines the application of SWAT model for streamflow simulation in an experimental basin with mixed-land-use characteristics (i.e., urban/peri-urban) using daily and hourly rainfall observations. The main objective of the present study was to investigate the influence of rainfall resolution on model performance in order to

analyze the mechanisms governing surface runoff at the catchment scale. The model was calibrated for 2018 and validated for 2019 using the SUFI-2 algorithm in the SWAT-CUP program. Daily surface runoff was estimated using the Curve Number method and hourly surface runoff was estimated using the Green and Ampt Mein Larson method. A sensitivity analysis conducted in this study showed that the parameters related to groundwater flow were more sensitive for daily time intervals and channel routing parameters were more influential for hourly time intervals. Model performance statistics and graphical techniques indicated that the daily model performed better than the sub-daily model (daily model: NSE = 0.86, $R^2$ = 0.87, PBIAS = 4.2%, sub-daily model: NSE = 0.6, $R^2$ = 0.63, PBIAS = 11.7%). The Curve Number method produced higher discharge peaks than the Green and Ampt Mein Larson method and estimated better the observed values. Overall, the general agreement between observations and simulations in both models suggests that the SWAT model appears to be a reliable tool to predict discharge in a mixed-land-use basin with high complexity and spatial distribution of input data".

We will make the following changes in the Introduction section in the revised manuscript in order to emphasize the objectives of the study:

**Lines 72-73**: "In this study, the SWAT 2012 model (rev 681) in the QSWAT interface was used to simulate streamflow in an experimental basin using daily and sub-daily (hourly) rainfall observations. The main objectives were to (i) calibrate and validate the SWAT model using streamflow data, (ii) examine which parameters are more sensitive in different time steps, (iii) estimate the influence of rainfall resolution on model performance, (iv) compare the Curve Number method and Green and Ampt Mein Larson method for runoff simulation, (v) examine the accuracy of the sub-daily model and compare the peak discharges and time of peak of the two models in selected rainfall events, and (vi) investigate the suitability of the SWAT model for hourly simulation in a mixed-land-use basin (i.e., blended combinations of land use). Hence, this study will provide essential hydrological knowledge and contribute to the understanding of the earth surface processes of an urban/peri-urban hydrological system with complex land use in order to analyze the mechanisms governing surface runoff at the catchment scale".

We will make the following changes in the Results and Discussion section in the revised manuscript:

[revised manuscript text omitted]

---

## Author Comment (AC3)

**Reply to comments of Anonymous Referee #3**

We thank anonymous Referee #3 for reviewing our manuscript. The authors are grateful for the insightful comments which provide great suggestions to improve the manuscript.

**General Comments**

The authors present a modeling study of an urban catchment with the SWAT. The study aims to compare the performance of daily simulations (using the curve number method) and hourly simulations (using the GAML method). The paper is within the scope of the journal, address relevant scientific questions and the presentation is of quality.

The Methods and assumptions are well described and understandable. However, Results and Discussion could be developed: at the moment they read a little like a list of comments (especially section 3.3) without enough physical discussion. The authors find that different modeling time steps and different runoff generation methods impact model parameters and performance, which is somehow expected and established. The novelty of the contribution therefore needs to be detailed and explained.

The reasons why the sub daily model did not perform as well as the daily model deserve more physical discussion about hydrological processes and implications. In addition, is it really fair to compare hourly and daily performance metrics directly? Shouldn't the results of the GAML method be aggregated to the daily timestep to really be compared with the CN method?

Reply: We thank the referee for the constructive feedback and comments. We agree that the Results and Discussion should be improved. We will re-organize Section 3.2 and Section 3.3 in a more clear way and we will discuss the physical meaning of the results in the revised manuscript. We will explain the objectives and the contribution of this study in Introduction, Results and Discussion section. In addition, we will explain the peak flow underestimation and we will discuss the disadvantages of the GAML method. We will also aggregate the hourly results to a daily time step in order to compare the performances in the revised manuscript. We will make the suggested changes in the following comments.

**Specific comments**

**Abstract**

Comment: The abstract does a good job at summarizing the work in concise and clear language. The novelty of study could be explained in the Abstract and numerical values of Results could also be given.

Reply: We thank the referee for the constructive feedback. We agree with the referee that the novelty of the study should be more emphasized. We will make the following changes in the revised manuscript:

**Lines 11-22**: "SWAT (Soil and Water Assessment Tool) is a continuous time, semi-distributed river basin model that has been widely used to evaluate the effects of alternative management decisions on water resources. This study examines the application of SWAT model for streamflow simulation in an experimental basin with mixed-land-use characteristics (i.e., urban/peri-urban) using daily and hourly rainfall observations. The main objective of the present study was to investigate the influence of rainfall resolution on model performance in order to analyze the mechanisms governing surface runoff at the catchment scale. The model was calibrated for 2018 and validated for 2019 using the SUFI-2 algorithm in the SWAT-CUP program. Daily surface runoff was estimated using the Curve Number method and hourly surface runoff was estimated using the Green and Ampt Mein Larson method. A sensitivity analysis conducted in this study showed that the parameters related to groundwater flow were more sensitive for daily time intervals and channel routing parameters were more influential for hourly time intervals. Model performance statistics and graphical techniques indicated that the daily model performed better than the sub-daily model (daily model: NSE = 0.86, $R^2$ = 0.87, PBIAS = 4.2%; sub-daily model: NSE = 0.6, $R^2$ = 0.63, PBIAS = 11.7%). The Curve Number method produced higher discharge peaks than the Green and Ampt Mein Larson method and estimated better the observed values. Overall, the general agreement between observations and simulations in both models suggests that the SWAT model appears to be a reliable tool to predict discharge in a mixed-land-use basin with high complexity and spatial distribution of input data".

Comment: L12: Is 'demonstrate' the right word? The paper rather "examines" the influence of precipitation time-step on performance metrics.

Reply: We agree with the referee. The verb "demonstrate" is not the right word. This study rather investigates or examines the influence of precipitation time step on model performance. We will make the following changes:

**Lines 12-14**: "This study examines the application of SWAT model for streamflow simulation in an experimental basin with mixed-land-use characteristics (i.e., urban/peri-urban) using daily and hourly rainfall observations".

Comment: L21: "long periods of time": in the paper the modeling period is 3 years which is not very long.

Reply: We agree with the referee that three years is not a long period of time. We believe that we should focus in the land use characteristics of the study area and in the ability of SWAT to predict discharge in an urban/peri-urban environment. We will make the following changes in the manuscript.

**Lines 21-22**: "Overall, the general agreement between observations and simulations in both models suggests that the SWAT model appears to be a reliable tool to predict discharge in a mixed-land-use basin with high complexity and spatial distribution of input data".

**1. Introduction:**
Comment: L74: water level data of water flow data?

Reply: We thank the referee for this comment. The river gauge at the outlet of the basin measures water level data at a 15 min time step. Then, using the Manning's equation we calculated the flow velocity and the flow rate.

Manning's Equation:

$$Q = V * A = \left(\frac{1}{n}\right) * A * R^{\frac{2}{3}} * S^{\frac{1}{2}}$$

Where:

Q = Flow Rate, (m$^3$/s)

V = Velocity, (m/s)

A = Flow Area, (m$^2$)

n = Manning's Roughness Coefficient

R = Hydraulic Radius, (m)

S = Channel Slope, (m/m)

**2. Materials and Methods:**
Comment: L95-96: "in different times and under different weather conditions": is the monitoring continuous?

Reply: The monitoring stations measure water level at a 15 minute time step and the data are provided freely from Open Hydrosystem Information Network (OpenHi.net).

Comment: L101-106: The authors could consider adding a sentence about velocity measurement? Can the real life accuracy of the probes be somehow estimated (especially since observational errors are mentioned later in the discussion)?

Reply: We thank the referee for this comment. The station at the outlet of the basin measures water level data. Then, using the Manning's equation we calculated the flow velocity and the flow rate. We will also exclude the reference to the water velocity in the revised manuscript since we didn't use water velocity in the calibration process. We will shorten the instruments description. We will clarify this statement in the manuscript in the following lines:

**Lines 95-106**: "The study area includes four water level monitoring stations that provide continuous recordings of the river stage at pre-selected time-intervals (15mins time-step) (Fig. 1). The stations were installed at the end of 2017 under the supervision of the School of Mining of National Technical University of Athens (NTUA). The network was developed under the EU H2020 RIA Program SCENT (Smart Toolbox for Engaging Citizens in a People-Centric Observation Web). The station which is located at the outlet of the study area was selected as the most suitable for further analysis in this study, because the three upstream stations experienced some mechanical problems that affected the calibration and validation process. The monitoring stations are part Open Hydrosystem Information Network (OpenHi.net) which is a national integrated information infrastructure for the collection, management and free dissemination of hydrological data (OpenHi.net) in Greece."

Comment: L133: and sub-daily?

Reply: We thank the referee for this comment. The SWAT model operates on a daily and sub-daily time step. We will make the following changes in the revised manuscript:

**Lines 133-134**: "The model operates on a daily time step, and it has been recently updated to sub-daily time step computations (Jeong et al., 2010). SWAT has been developed to evaluate the impact of management practices on water, sediment, and agricultural chemical yields in large river basins over long time periods."

Comment: L198: mean and standard deviation of daily discharge? In $m^3/s$?

Reply: We thank the referee for this comment. The numbers refer to discharge. We changed line 198 in the following way:

**Line 198**: "Mean and standard deviation of discharge for 2018 were 1.25 and 0.46 and for 2019 were 1.42 and 0.74 respectively".

**3. Results and Discussion**

**Section 3.1**

This section describes the results of the sensitivity analysis. It shows that the daily simulation seems to be more sensitive to runoff generation parameters whereas the sub-daily simulation is more sensitive to channel routing parameters. The section could be better structured: for example it starts with a discussion on CH_N2, then discuss GWQMN and GW_REVAP, then again CH_N2 and in the same paragraph discuss CN2, which makes it hard for the reader to follow the reasoning.

Reply: We thank the referee for the suggestions and we agree that this section should be better structured. We joined the first two paragraphs, we deleted the repetition in lines 241-242 and we explained the difference in the calibrated values of the two models. We will make the following changes in the revised manuscript in Section 3.1:

**Lines 231-257**: "The most sensitive parameters obtained in daily and hourly simulation are presented in Table 4. Sensitive parameters are characterized by large t-Test and small p-Value. The parameters were characterized as significantly sensitive when the p-value was less than 0.03. In the daily model, the most sensitive parameters were deep aquifer percolation fraction (RCHRG_DP), groundwater delay time (GW_DELAY), lateral flow travel time (LAT_TTIME), average slope steepness (HRU_SLP) and moist bulk density (SOL_BD). These parameters were connected to groundwater flow, runoff generation and channel routing. In the sub-daily model, the significantly sensitive parameters were average slope steepness (HRU_SLP), Manning's "n" value for the main channel (CH_N2), effective hydraulic conductivity in main channel alluvium (CH_K2) and lateral flow travel time (LAT_TTIME). These were all related to channel routing.

The differences in the sensitivity of the calibrated parameters of the two models reflect the impact of the operational time step on model performance (Boithias et al., 2017; Jeong et al., 2010). In particular, the hourly model is characterized by larger GWQMN and GW_REVAP values than the daily model. GWQMN is the threshold depth of water in the shallow aquifer required for return flow to occur and GW_REVAP controls the water movement from the shallow aquifer into the overlying unsaturated soil layers. As these parameters increase, the rate of evaporation increases up to the rate of potential evapotranspiration, resulting in a corresponding decrease of the baseflow. Furthermore, the fitted value of CH_N2 in the hourly simulation was $0.11(m^{-1/3}s)$ and was larger than $0.08\ (m^{-1/3}s)$ in the daily simulation. The CH_N2 parameter affects the rate and the velocity of flow (Boithias et al., 2017). Therefore, the larger CH_N2 value was connected to smaller flow velocity. According to Boithias et al. (2017), the CH_N2 parameter is more sensitive at the hourly time step rather than the daily time step, because at the daily time step the flow peak is influenced by other processes decreasing the sensitivity of the CH_N2. In addition, the value range for CN2 was smaller for the sub-daily model, leading thereby to lower peak flows. Other differences were average slope steepness

(HRU_SLP), average slope length (SLSUBBSN), groundwater delay time (GW_DELAY) and Manning's "n" value for overland flow (OV_N). Their values were all smaller in sub-daily simulation. Overall, the differences between the two models lay mostly in the different runoff estimation methods used by the two models."

Comment: L241-242: repetition from lines 234-239.

Reply: We agree with the referee. We deleted the repetition from lines 234-239 as mentioned above.

Comment: L252: is this difference physically meaningful?

Reply: Channel and hillslope velocities define the time of the peak and the shape of the hydrograph. In the SWAT model, the CH_N2 parameter (Manning's "n" value for the main channel) affects the rate and the velocity of the flow (Boithias et al., 2017). In this study, the CH_N2 parameter showed the highest sensitivity in the hourly model. This outcome is similar to those of previous studies (Boithias et al., 2017; Jeong et al., 2010). In particular, according to Boithias et al. (2017) the CH_N2 parameter is more sensitive at the hourly time step rather than the daily time step, because at the daily time step the flow peak is influenced by other processes decreasing the sensitivity of the CH_N2. Therefore, the lower peak flows of the hourly model comparing to the daily model could be attributed to the different calibrated value of the CH_N2 parameter.

Comment: L253: CH_N2 or CN2?

Reply: It is CN2 (Curve Number) parameter. This parameter explains the reason why the sub-daily model has lower peak flows than the daily model.

**Section 3.2**

This section presents the performance metrics of the daily and subdaily simulations. The authors conclude that the CN method is better than the GAML method. But, as stated above, is it fair to compare daily simulations with hourly simulations? Shouldn't the hourly simulation be aggregated to a daily timestep to have a 'fair' comparison?

Reply: We thank the referee for the suggestion. We will aggregate the hourly results to a daily time step in order to compare the performances in the revised manuscript. We present below in Figure 3 the observed versus simulated daily discharge aggregated from hourly outputs during the calibration and validation processes. We also show in Figure 4 the flow duration curve for the daily discharge aggregated from hourly outputs. We present the statistics for the daily aggregated discharge in Table 5. We will include these changes in the revised manuscript.

Comment: L272: an explanation for the underestimation?

Reply: The GAML method requires detailed soil data and high resolution precipitation data which can be difficult to obtain and may affect the accuracy of the model's results. We will explain the peak flow underestimation and we will discuss the disadvantages of the GAML method in lines 295-299 in the following way in the revised manuscript:

**Lines 295-299**: "In this study, the daily model produced higher discharge peaks than the hourly model and generally estimated better the observed values. These results could be due to drawbacks of the GAML method, such as the requirement for detailed soil information and high resolution rainfall data in a sub-daily time step (King et al., 1999). The GAML method assumes that the soil profile is characterized by homogeneity and that the previous soil moisture is distributed uniformly in the soil profile (Jeong et al., 2010). Therefore, the uncertainty in the resolution of the rainfall data, the heterogeneity of the soil formations and the upcoming difficulty in determining the parameters' values for parameterization could affect the method's efficiency. The selection of sub-daily precipitation input time step as well as the resolution of the precipitation data have a great impact on model results when using the GAML method and it should be based on the scale and characteristics of the watershed (Bauwe et al., 2016; Jeong et al., 2010; Kannan et al., 2007). Furthermore, observational errors in the model input data (i.e., weather, soil and land use data) include inaccuracies in the estimation of channel and hillslope velocities and channel geometry, in the nature of the sensor, environmental conditions and data collection (Guzman et al., 2015). These errors can generate variability, lead to undesired trends, and influence the model calibration and validation results (Kamali et al., 2017). In addition, the complex land use characteristics and processes of an urban/peri-urban environment and assumptions made during the model structure/parameterization process (e.g., selection of parameters for calibration, objective function, and conceptual simplifications) increase the uncertainty of the results."

Comment: L274: it is expected that a daily timestep performs better than a subdaily timestep. It could be interesting to compare both methods at the same timestep, as stated above.

Reply: We agree with the referee and we will aggregate the hourly results to a daily time step in order to compare the performances in the revised manuscript. We will make the following changes in the revised manuscript:

**Lines 266-277**: "Quantitative statistics and criteria recommended by Moriasi et al. (2007, 2015) were used to evaluate the model performance. In order to investigate the influence of rainfall on model performance and compare daily outputs to hourly outputs, the hourly outputs were aggregated to daily averages. Figure 2 shows the temporal dynamics of the hydrographs reproduced by both infiltration methods. The high flow season is observed during winter and spring.  The low flow season is observed in summer and early fall due to high evapotranspiration. Figure 3 shows the observed versus the simulated daily discharge aggregated from hourly outputs during the calibration and validation processes. Figure 4 presents the flow duration curves of the models, indicating good agreement between observed and simulated values. Generally, in the sub-daily model, the simulated discharge peaks did not always match the observed values and were sometimes considerably lower.

The performance statistics are illustrated in Table 5 and indicate reasonable calibrated models for both infiltration approaches. Model performance using the CN method showed better results than the GAML method. In particular, the NSE and $R^2$ indices for the daily model were 0.84 and 0.79 for the calibration period and 0.87 and 0.86 for the validation period. For the sub-daily model the NSE and $R^2$ indices were 0.53 and 0.49 for the calibration period and 0.63 and 0.6 for the validation period respectively. In addition, when the hourly outputs were aggregated to daily averages the NSE was improved comparing the NSE of the sub-daily model (e.g., sub-daily model: $NSE_{calibration} = 0.49$ and $NSE_{validation} = 0.6$, daily averages: $NSE_{calibration} = 0.66$ and $NSE_{validation} = 0.78$). However, the daily model outperformed the daily aggregated discharge during both calibration and validation periods. Furthermore, the daily model showed smaller modeling uncertainties with P-factor 0.79 and R-factor 1.58 (compared to 0.83 and 1.71 respectively for the sub-daily model)."

**Table 5. Model evaluation statistics of the daily, sub-daily and daily aggregated from hourly outputs SWAT models for the calibration and validation periods.**

| Time-step | Period | p-Factor | r-Factor | $R^2$ | NSE | PBIAS(%) |
|---|---|---|---|---|---|---|
| Daily | Calibration | 0.74 | 1.41 | 0.84 | 0.79 | 6.4 |
| | Validation | 0.79 | 1.58 | 0.87 | 0.86 | 4.2 |
| Sub-Daily | Calibration | 0.72 | 1.33 | 0.53 | 0.49 | 16.9 |
| | Validation | 0.83 | 1.71 | 0.63 | 0.6 | 11.7 |
| Daily averages | Calibration | - | - | 0.76 | 0.66 | 16.8 |
| | Validation | - | - | 0.82 | 0.78 | 11.6 |

[Figure]

**Figure 3. Observed and simulated daily discharge (m³ s⁻¹) aggregated from hourly outputs: calibration period (a) and validation period (b).**

[Figure]

**Figure 4. Observed and simulated flow duration curves (m³ s⁻¹) at the daily time step (a), at the hourly time step (b), and at the daily discharge aggregated from hourly outputs time step (c).**

Comment: L284: performance metrics are satisfactory, but performance metrics also depend on what we want to use the model for: for example, though the model here replicates the timeseries quite well, it could not be trusted for flood analysis (poor performance on hourly peaks).

Reply: We thank the referee for this comment. This study aims to investigate the complex hydrological processes that take place in a mixed-land-use basin in order to understand the mechanisms that create surface runoff. We agree with the referee that the aims and novelty of this study should be more emphasized. We will make the following changes in the Introduction section in the revised manuscript:

Lines 72-73: "In this study, the SWAT 2012 model (rev 681) in the QSWAT interface was used to simulate streamflow in an experimental basin using daily and sub-daily (hourly) rainfall observations. The main objectives were to (i) calibrate and validate the SWAT model using streamflow data, (ii) examine which parameters are more sensitive in different time steps, (iii) estimate the influence of rainfall resolution on model performance, (iv) compare the Curve Number method and Green and Ampt Mein Larson method for runoff simulation, (v) examine the accuracy of the sub-daily model and compare the peak discharges and time of peak of the two models in selected rainfall events, and (vi) investigate the suitability of the SWAT model for hourly simulation in a mixed-land-use basin (i.e., blended combinations of land use). Hence, this study will provide essential hydrological knowledge and contribute to the understanding of the earth surface processes of an urban/peri-urban hydrological system with complex land use in order to analyze the mechanisms governing surface runoff at the catchment scale".

Comment: L289: what is 'ET runoff generation'?

Reply: We thank the reviewer for this comment. Kannan et al. (2007) identified a suitable combination of evapotranspiration (i.e., Penman-Monteith, Hargreaves) and infiltration (i.e., Curve Number, GAML) methods for runoff generation. We believe that the line 289 is not necessary. We will make the following changes in lines 289-293:

Lines 289-293: "Kannan et al. (2007) conducted a sensitivity analysis to identify the best combination of evapotranspiration and infiltration method for runoff generation and concluded that the CN method performed better than the GAML method for streamflow because the GAML method tends to hold more water in the soil profile and predict a lower peak runoff rate".

Comment: L296: This is interesting but a little unclear: what is meant by 'too large'? Would the results be better with, say, 10 min rainfall? Why?

Reply: We thank the referee for highlighting this point. As we mentioned above the GAML method comparing to the CN method requires detailed soil data and precipitation data of high resolution. We wanted to emphasize the impact of rainfall resolution and accuracy on model performance. The selection of sub-daily precipitation time step is very crucial when using the

GAML method and it should be based according to the characteristics of each basin. We will rewrite the manuscript to be as follows:

**Lines 295-299**: "In this study, the daily model produced higher discharge peaks than the hourly model and generally estimated better the observed values. These results could be due to drawbacks of the GAML method, such as the requirement for detailed soil information and high resolution rainfall data in a sub-daily time step (King et al., 1999). The GAML method assumes that the soil profile is characterized by homogeneity and that the previous soil moisture is distributed uniformly in the soil profile (Jeong et al., 2010). Therefore, the uncertainty in the resolution of the rainfall data, the heterogeneity of the soil formations and the upcoming difficulty in determining the parameters' values for parameterization could affect the method's efficiency. The selection of sub-daily precipitation input time step as well as the resolution of the precipitation data have a great impact on model results when using the GAML method and it should be based on the scale and characteristics of the watershed (Bauwe et al., 2016; Jeong et al., 2010; Kannan et al., 2007). Furthermore, observational errors in the model input data (i.e., weather, soil and land use data) include inaccuracies in the estimation of channel and hillslope velocities and channel geometry, in the nature of the sensor, environmental conditions and data collection (Guzman et al., 2015). These errors can generate variability, lead to undesired trends, and influence the model calibration and validation results (Kamali et al., 2017). In addition, the complex land use characteristics and processes of an urban/peri-urban environment and assumptions made during the model structure/parameterization process (e.g., selection of parameters for calibration, objective function, and conceptual simplifications) increase the uncertainty of the results."

**Section 3.3**

This section focuses on six "heavy" rainfall events in which the authors describe, in text, peak flow values and average flow values during the events. The hourly model underestimates peak flows. This section comes as a surprise for the reader as it is not mentioned in the Methods. Moreover, it is unclear how the events were selected: are they the 6 larger events of the timeries? It would be interesting to have an estimation of their return period to define "heavy"? From line 312 to line 335, the text simply describes hydrographs, without comments or analysis. Maybe the authors could consider a Table instead with rainfall characterstics (totals, duration, return period, etc.) and standard describers of hydrographs (peaks flows, difference between peaks, etc.)?

Reply: We thank the referee for the suggestions. We agree that Section 3.3 is not mentioned in the Methods. Therefore, we will mention this section in the Introduction in lines 72-73. We will discuss the model performance in Section 3.2 and we will discuss the selected rainfall events in Section 3.3. We also agree that the term "heavy" should be more explained. The rainfall events that were selected were the events with the highest rainfall intensity, discharge and return period. 
[revised manuscript text omitted]

| 12/01-14/01/2018 | 2.6 | 10.7 | 6:00 | 2.2 | 7.1 | 5:00 |
| 05/05-07/05/2018 | 2.2 | 11.1 | 20:00 | 2.1 | 6.6 | 21:00 |
| 29/09-01/10/2018 | 5.7 | 17.2 | 18:00 | 5.2 | 8.9 | 18:00 |
| 05/02-07/02/2019 | 3.6 | 16.2 | 1:00 | 2.9 | 6.2 | 00:00 |
| 12/11-14/11/2019 | 2.9 | 12.3 | 3:00 | 2.4 | 3.5 | 3:00 |
| 29/12-31/12/2019 | 4.9 | 14.8 | 21:00 | 3.6 | 6.9 | 21:00 |

It is concluded that the underestimation of peak flows is due to uncertainty in observed data or input data (rainfall), but without many arguments or any estimation of these uncertainties. How can one be sure that the errors are due to the data and not to the model? There probably exists a parametrization which can replicate high flows much better, with poorer performance on low flows?

Reply: We thank the referee for highlighting this point. The issue of uncertainty is a major topic in hydrological modeling and has been discussed in many publications (Harmel et al., 2006, 2014; Kamali et al., 2017; Kouchi et al., 2017; Tan et al., 2020). In general, model errors are due to inaccuracies (i) in the quality of input data (i.e., weather, soil and land use data), (ii) in the conceptual model, (iii) in the choice of objective function, (iv) the observed data and (v) in the parameterization. The correct selection and combination of the values of the parameters that influence surface runoff, groundwater, channel routing and evapotranspiration is a critical point in model calibration (Polanco et al., 2017). There are many possible combinations of parameters that can replicate high flows much better but in every case the ranges of the calibrating parameters should be kept in reasonable limits using quantitative statistics and graphical comparisons in order to ensure that hydrological processes represent the characteristic of the study area (Daggupati et al., 2015). In this study, the initial ranges of the calibrating parameters were set according to literature and sensitivity analysis. Then, based on the performance of the default model, specific parameters were parameterized using calibration protocols (Abbaspour et al., 2015). We will emphasize in the revised manuscript the many sources of uncertainty that exist in this study and we will discuss them in the following comment.

Comment: L336-340: It is correct that errors in a model can be explained by 1/ uncertainty in the observed data 2/ uncertainty in input data or 3/ the model structure/parameterization. But what about this particular study? This could be further discussed.

Reply: We thank the referee for this comment. The values of the calibrated parameters and their sensitivities are influenced by the type and quality of input data, the conceptual model, the choice of the objective function and inaccuracies in measured input data used for calibration and validation (Abbaspour et al., 2015; Arnold et al., 2012; Polanco et al., 2017). In this study the errors can be explained by the uncertainty of the input data (e.g., quality of the precipitation data, land use data, observed discharge data, Manning's equation for flow estimation), the complex land use characteristics of an urban/peri-urban environment which are difficult to simulate in SWAT, and the differences behind the mechanisms of the CN method and the GAML method for streamflow estimation. In addition, assumptions made during the model structure/parameterization process (e.g., selection of parameters for calibration, objective function, and conceptual simplifications) increase the uncertainty of the results.

We will make the following changes in the revised manuscript in lines 336-340:

[revised manuscript text omitted]

4. Conclusions:

Comment: L366: 3 years is not really "long time".

Reply: We agree with the referee that three years is not a long period of time. We believe that we should focus in the land use characteristics of the study area and in the ability of SWAT to predict discharge in an urban/peri-urban environment. We will make the following changes in the manuscript in line 366.

**Line 366**: "Overall, the general agreement between observations and simulations in both models suggests that the SWAT model appears to be a reliable tool to predict discharge in a mixed-land-use basin with high complexity and spatial distribution of input data."

Comments on Figures:

Comment: Figure 1 and Figure 2 could be merged

Reply: We agree with the referee. We merged Figure 1 and Figure 2. We will add the following figure in the revised manuscript:

[Figure]

**Figure 1. Geographical location of the study area (a) and spatial distribution of land use (b) and soil (c).**

We will also make the following changes in the manuscript in lines 80-87:

**Lines 80-87**: "The study area includes the upper part (NW sub-basin) of the Kifissos River basin, located in Athens Greece (Fig. 1a). The Kifissos River basin occupies an area of 380 km² and its route is approximately 22 km, of which at least 14 km are within an urban area. The elevation ranges from 94 m to 1399 m with plains in the south and hills in the north part of the basin. The mean annual temperature is 16.4 °C and the mean annual rainfall across the basin is 577.2 mm.

The study area is characterized as an urban/sub-urban area, with residential areas, shrubland and agriculture accounting for 34.1, 15.9 and 12.4 % of its land use coverage, respectively (Fig. 1b). It includes mainly four soil types, Cambisols, Regosols, Leptosols and Luvisols (Fig. 1c). The dominant soil formations are characterized by good soil permeability and high contents of clay and sand."

Comment: Figure 3: It could be worth to add rainfall?

Reply: We agree with the referee. We added rainfall in Figure 3. We will add the following figure in the revised manuscript:

[Figure]

**Figure 2. Observed and simulated discharge (m$^3$ s$^{-1}$) at the daily time step (a, b) and at the hourly time step (c, d).**

**Minor comments and typos**

Comment: L12: this study demonstrates (remove the comma)

Reply: We agree with the referee. We will make the following changes in lines 12-14:

**Lines 12-14**: "This study examines the application of SWAT model for streamflow simulation in an experimental basin with mixed-land-use characteristics (i.e., urban/peri-urban) using daily and hourly rainfall observations."

Comment: L29: they are 'used to monitor', not 'able'

Reply: We agree with the referee. We will make the following changes in lines 29-30:

**Lines 29-30**: "They are also used to monitor the major components of the surface hydrological cycle by using remote sensing and geophysical measurements (Tauro et al., 2018)."

Comment: L30: they monitor groundwater

Reply: We agree with the referee. We will make the following changes in lines 30-31:

**Lines 30-31**: "Furthermore, they monitor groundwater and river water quality with the use of tracer experiments which can estimate the residence and travel times of water in different components of the hydrological cycle  (Hrachowitz et al., 2016; Stockinger et al., 2016)."

Comment: L81: "route" is unclear

Reply: We thank the referee for this comment. We will rephrase line 81 in the following way:

**Line 81: "**The Kifissos River basin occupies an area of 380 km$^2$. Kifissos River route is approximately 22 km, of which at least 14 km are within an urban area."

**References**

[revised manuscript text omitted]

---

## Author Response (AR2)

National Technical University of Athens
School of Mining & Metallurgical Engineering
Laboratory of Engineering Geology & Hydrogeology

**Article Cover letter**

Dear Editor,

We would like to submit the revised version of our paper "Hydrological modeling using the SWAT Model in urban and peri-urban environments: The case of Kifissos experimental sub-basin (Athens, Greece)" by Evgenia Koltsida, Nikos Mamassis, and Andreas Kallioras.

We believe we emphasized the novelty of our study and its potential impact and that we followed all the proposed corrections by the reviewer that substantially improved the article. We are looking forward to hearing from you concerning the review process. We would also like to thank Ref. #1 for accepting our manuscript as is.

We confirm that this manuscript has not been published elsewhere and is not under consideration by another journal. The authors have no conflicts of interest to disclose and have all approved this submission.

Thank you for your time and consideration.

Sincerely,
Evgenia Koltsida, M.Sc.

**Reply to comments of Referee #2**

We thank Referee #2 for reviewing our manuscript and for giving comments and suggestions. Our answers are given in blue, below, while the original text of the review was kept in black.

**Comments**

**General comment**

The authors have evaluated the impacts of different precipitation resolution (daily and sub-daily) and runoff simulation methods (Curve Number and GAML) on SWAT simulations in the upper part of the Kifissos River Basin, Athens Greece. The topic is interesting, but the presentation of results (figures and tables) and discussion needs to be improved. Besides that, the contribution of this study is marginal and not up to the HESS's standard. I can't really find any new innovative elements in the manuscript. I personally feel that this manuscript is more suitable for a normal journal rather than HESS, unless the authors make a major revision by adding some new contributions, i.e., testing conditions or elements that have yet to consider in previous studies, development a better framework in characterize the uncertainty of methods or inputs in SWAT modelling, or improvement of SWAT in simulation of urban and peri-urban environment.

Reply: We thank the referee for the constructive feedback and comments. We will re-organize the Introduction and Results and Discussion sections. We will explain the study's objectives and novelty and emphasize the importance of the specific study area in the Introduction section. In addition, we will discuss the sources of uncertainty more accurately in the Results and Discussion section. We will also explain the figures, and we will improve the writing of the revised manuscript. We will make the suggested changes in the following comments.

Please find my comments as follows:

**Comment #1**

Line 71: I would expect to see the usage of the SWAT+ model since it has already been released last year.

Reply: We thank the referee for the suggestion. The option for sub-daily simulation in SWAT+ model is currently being worked on and should be available hopefully in the next release. We would like to test the model in the near future.

**Comment #2**

Lines 72 – 77: Too many objectives. Besides that, I am unable to see the novelty of this study. The comparisons of different precipitation resolution and runoff simulation methods on SWAT simulations have been widely conducted by previous studies as stated in lines 53-70, it seems like the authors just repeat the same thing. Ideally, the authors should propose a new framework or method to better quantify the uncertainties.

Reply: We thank the referee for the insightful comments and suggestions. We agree with the referee that the aims and novelty of this study should be emphasized more. Therefore, we will reorganize the Introduction and Results sections and discuss the many sources of uncertainty in this study in the revised manuscript. It is worth noting that this study aims to understand and estimate the hydrological components of the area and will establish a basis for further modeling applications. We will make the following changes in the revised manuscript:

**Lines 72-77:** "The selected study area has been severely urbanized from 1990 till today, at the expense of forests and agricultural areas. During this period, the artificial surfaces increased by 69.93%, and the agricultural areas and the forests decreased by 54.14% and 14.34%, respectively (Corine Land Cover, CLC 1990-2018). The area is portrayed as an urban/peri-urban system with about 51% of artificial surfaces, 13% of agricultural areas, and 36% of forests and semi-natural areas. The interaction between different land uses (e.g., urban and rural characteristics) contributes to the formation of a complex environment characterized by high variability in

management practices, rapid response, and diverse hydrological processes, which may increase problems of model uncertainties (Boithias et al., 2017). Land use maps and soil maps may not capture this complex environment precisely, enhancing the SWAT model's difficulty in representing and simulating the actual conditions of the basin, which can affect water discharge. In addition, the study area is a typical Mediterranean catchment, vulnerable to natural hazards (i.e., flash floods and forest fires). In order to interpret the behavior of such a complex environment, the SWAT 2012 hydrological model (rev 681) was used for its realistic representation. The available studies that used the sub-daily option of the SWAT model refer mainly to agricultural (Bauwe et al., 2016; Boithias et al., 2017; Cheng et al., 2016; Ficklin and Zhang, 2013; Golmohammadi et al., 2017; Maharjan et al., 2013; Yang et al., 2016; Yu et al., 2018) or small urban catchments (Campbell et al., 2018; Jeong et al., 2010; Li and DeLiberty, 2020) and rarely in peri-urban catchments. Thus, the suitability of the sub-daily option of the SWAT model to simulate discharge in a peri-urban catchment has not been extensively tested. The main objectives were (i) to investigate which parameters are more sensitive under different temporal time steps in a mixed-land-use basin (i.e., blended combinations of land use, management practices, and hydrological processes), (ii) to compare the results of the hourly time-step simulation (Green and Ampt Mein Larson method) to those obtained from daily time-step simulation (Curve Number method) and (iii) evaluate the sub-daily option of the SWAT model for discharge simulation by examining peak discharges and time of the peak of selected rainfall events. The calibration methodology developed in this catchment can be applied to areas with similar hydrological-meteorological and geomorphological attributes (i.e., Mediterranean peri-urban areas). This study provides essential hydrological knowledge and contributes to understanding the critical processes of an urban/peri-urban system to analyze the mechanisms governing surface runoff at the catchment scale. The outcomes will establish a basis for further modeling applications, which will be helpful for local planners to use in future regional urban development strategies. The study area information, methodology, and data input are presented in Section 2, results and discussions are detailed in Section 3, and conclusion is provided in Section 4."

**Comment #3**

Line 110: What is the accuracy of the 2018 Corine land cover map?

Reply: The mapping methodology used in CLC 2018 was the same as in CLC 2012. More than 15 countries larger than 90.000 km$^2$ were evaluated separately, while countries smaller than 90.000 km$^2$ were grouped together (7 groups). Independent experts evaluated more than 25.000 sampling locations. Overall, the results showed that the 85% target accuracy has been achieved; however some differences between countries exist, due to geographic complexity. Table 1 presents the characteristics of the CLC 2018.

**Table 1. Corine land cover characteristics (Corine Land Cover, 2018).**

| Product characteristics | Corine Land Cover (CLC) 2018 |
|---|---|
| Satellite data | Sentinel-2 and Landsat-8 for gap filling |
| Time consistency | 2017-2018 |
| Geometric accuracy, satellite data | ≤ 10 m (Sentinel-2) |
| Min. mapping unit/width | 25 ha / 100 m |
| Geometric accuracy, CLC | better than 100 m |
| Thematic accuracy, CLC | ≥ 85% |
| Change mapping (CHA) | boundary displacement min.100 m; all changes ≥ 5 ha are to be mapped |
| Thematic accuracy, CHA | ≥ 85% |
| Production time | 1.5 years |
| Documentation | standard metadata |
| Access to the data (CLC, CHA) | free access for all users |
| Number of countries involved | 39 |

**Comment #4**

Lines 115-118: I would suggest the authors to collect up-to-date hydro-climatic data for longer calibration and validation periods.

Reply: We thank the referee for the suggestion. The monitoring station was installed at the end of 2017 and provided discharge data from 01/01/2018 to 31/12/2019. Before this date, there were no discharge measurements, and the estimation of the water balance components of the area was not feasible. We know that the time period is not long, but there were no observed discharge data in this area. Therefore, due to data availability, we split the data from 01/01/2018 to 31/12/2018 for calibration and from 01/01/2019 to 31/12/2019 for validation. In the future, we intend to update the calibration and validation periods when more data becomes available.

**Comment #5**

Lines 163-164: It would be better to conduct calibration from 2018 to 2019 and validation from 2020 to 2021.

Reply: We agree with the referee. In the future, we intend to update the calibration and validation periods when more data becomes available (See also Comment #4).

**Comment #6**

Lines 191-193: This sentence has been described in lines 163-165. Can consider to remove it.

Reply: We agree with the referee. We will remove this sentence in the revised manuscript.

**Comment #7**

Line 263: Is the "infiltration" correct term? Suppose methods for runoff simulations. Right?

Reply: We agree with the referee, we will make the following changes in the revised manuscript.

**Line 263:** "Figure 2a and Figure 2b show the temporal dynamics of the hydrographs reproduced by both runoff estimation methods."

**Comment #8**

Lines 269-273: This part should describe about the differences in statistical values between CN and GAML methods. Not only the results for the CN method.

Reply: Lines 269-273 describe the performance statistics for the daily model, the hourly model, and the aggregated hourly to daily model. The main difference between the daily and the hourly model in SWAT is that surface runoff is estimated using the CN method in the daily model and the GAML method in the hourly model. Hence, by describing the differences between the daily and the hourly model, we also compared the performances of the CN and GAML method, respectively. We thank the referee, and we will clarify the differences between the two models in the revised manuscript.

**Comment #9**

Line 285 Figure 2: Need to be more specific when labelling each sub-figure. For example, I am not sure sub-figure (a) refer to what condition, calibration, validation or any infiltration methods.

Reply: We agree with the referee. We will update the Figure caption. We merged Figures 2 and 3. We will make the following changes in the revised manuscript:

[Figure]

**Figure 1. Observed and simulated discharge (m³s⁻¹) during calibration and validation periods: daily time step (a), hourly time step (b), and daily time step aggregated from hourly outputs (c). The CN method with daily rainfall observations was used for the daily model and the GAML method with hourly rainfall observations was used for the hourly model.**

**Comment #10**

Line 285 Figure 2(c, d): It seems like SWAT underestimated peak flows simulated under hourly scale. Any justification for this situation?

Reply: Figure 2 (c, d) (Figure 2b in the revised manuscript) shows that the hourly model produced lower peak flows than the daily model. We believe these results could be explained by disadvantages of the GAML method, errors in the input data, uncertainties in the observed data, and characteristics of the study area.

We will make the following changes in Results and Discussion section.

3.3 Comparison of selected rainfall events.

**Lines 301-308:** "Figure 4 shows the hydrographs of selected high rainfall events that occurred in the years 2018 and 2019 (Tatoi station, Lagouvardos et al., 2017). According to the study area's intensity-duration-frequency (IDF) curves the approximate return period of the selected episodes was ten years (T=10 years). These events were investigated to examine the accuracy of the sub-daily model and to compare the peak discharges and time of the peak of the two models. Table 6 displays the rainfall characteristics of each event (i.e., peak discharge, time of peak, and average discharge).

Generally, the hourly model underestimated the peak flows with values much lower than the observations for the majority of the events. These results confirm that the daily model using the CN method estimated better the observed values than the hourly model using the GAML method and was able to estimate with greater accuracy the peak discharge in most of the events. In addition, Figure 5 (a-c) shows that the discharge simulation improved after the main peak discharge event, especially in the last peaks. The improvement of the simulation as the rainfall events progressed indicates that the simulation requires time to adapt to the hydrological processes and conditions of the catchment (e.g., antecedent soil moisture conditions). These outcomes are similar to those of previous studies, which concluded that the best sub-daily performance for streamflow simulation appeared in wet antecedent soil moisture conditions and

suggested that the GAML method needs to improve the equation for infiltration routine (Jeong et al., 2010; Kannan et al., 2007; Meaurio et al., 2021)."

We will also revise the Discussion and Conclusion sections in the following comments.

**Comment #11**

Line 296 Table 5: The authors should include the statistical results for both the CN and GAML simulations.

Reply: Table 5 presents the statistics of the daily, sub-daily, and daily aggregated from hourly outputs SWAT models for the calibration and validation periods. The CN method with daily rainfall observations was used for the daily model, and the GAML method with hourly rainfall observations was used for the hourly model. The statistical results of the daily and hourly model reflect the performance of the CN and GAML method, respectively. We thank the referee and will emphasize the difference between the two models in the revised manuscript.

**Comment #12**

Lines 303-304: A comparison of the simulated flows under both the CN and GAML methods should be displayed in figures. In short, the quality of figures need to be improved.

Reply: The CN method with daily rainfall observations was used for the daily model and the GAML method with hourly rainfall observations was used for the hourly model. Therefore, Figure 4 shows the hydrographs of selected high rainfall events that occurred in the years 2018 and 2019 using the GAML method. We will clarify this in the Figure caption.

**Line 303-304:** "Figure 2. Observed and simulated hourly discharge ($m^3$ $s^{-1}$) using the GAML method for the heavy rainfall events that occurred in 2018 and 2019: (a) event from 12/01-14/01/2018; (b) event from 05/05-07/05/2018; (c) event from 29/09-01/10/2018; (d) event from 05/02-07/02/2019; (e) event from 12/11-14/11/2019; (f) event from 29/12-31/12/2019."

**Comment #13**

Lines 308-318: As mentioned earlier, many studies have compared the performance of CN and GAML in SWAT simulations and concluded that the CN method had a better performance. The authors only reported the same finding that already been confirmed in literature. So, what is the contribution of this study to the current knowledge on this topic?

Reply: The available studies that used the sub-daily option of the SWAT model refer mostly to agricultural (Bauwe et al., 2016; Boithias et al., 2017; Cheng et al., 2016; Ficklin and Zhang, 2013; Golmohammadi et al., 2017; Maharjan et al., 2013; Yang et al., 2016; Yu et al., 2018) or small urban catchments (Campbell et al., 2018; Jeong et al., 2010; Li and DeLiberty, 2020) and rarely in peri-urban catchments. Therefore, the suitability of the sub-daily option of the SWAT model to simulate discharge in a peri-urban catchment has not been extensively tested. Brighenti et al. (2019) concluded that the requirement of high-resolution input data, which is necessary for the sub-daily simulation, is the main challenge. In general, we believe that the application of the SWAT sub-daily option is a critical topic, and it is limited, especially in peri-urban catchments. Sub-daily calibration and validation are also limited; therefore, we believe this study will help other SWAT users with similar case studies to choose the most sensitive parameters and provide them with a calibration methodology. The outcomes will establish a basis for further modeling applications, which will be helpful for local planners to use in future regional urban development strategies. Overall, the results of our study suggest that the hourly option of the SWAT still needs improvement in timing-related routines and antecedent soil moisture conditions estimation. In the future, when more data becomes available, more events will be analyzed in order to investigate a correlation between antecedent moisture conditions and hourly model performance.

**Comment #14**

Line 320: Since the GAML method had a poorer performance than the CN method, I was wondering why the authors still applied the GAML method to assess the influence of daily and hourly scales precipitation data in SWAT modelling.

Reply: We thank the referee for highlighting this issue. Modeling results are very useful for managers and stakeholders for urban planning management. However, results at a daily or

monthly time step may not capture accurately the intensity of precipitation and the distribution of the hydrological processes. The CN method which is used for daily models does not account for rainfall intensity and duration, only total rainfall volume (King et al., 1999). Therefore, results at a sub-daily time step may be sufficient to capture the rapid response of peri-urban catchments. The CN method with daily precipitation data inputs was used for the daily model and the GAML method with sub-daily (hourly) precipitation data inputs was used for the sub-daily (hourly) model. We selected the SWAT model (Arnold et al., 2012) because it can perform long-term continuous simulation at a sub-daily time step and can also simulate single events which are important to understand and analyze runoff generation at the catchment scale. Since it first release, the SWAT model has been improved with algorithms for streamflow, erosion and sediment simulation (Jeong et al., 2014, 2010). In the near future, the new version SWAT+ will include algorithms for continuous sub-daily simulation and we would like to test its suitability (Bieger et al., 2017).

**Comment #15**

Line 331: I would expect to read more about the urban/peri-urban environment as stated in title and objective, but the description is relatively limited in the manuscript. One of the novelties that the authors can consider is the improvement or modification of SWAT simulations in urban/peri-urban environment. Otherwise, I don't think this manuscript is suitable to be published in HESS.

Reply: We thank the referee for the suggestions. We will present the many sources of uncertainty that exist in the study area, explain the results, and emphasize the study's importance in the Introduction, Results& Discussion, and Conclusion sections. We will make the following changes in the Results& Discussion and Conclusion sections.

**Results and Discussion**

3.3 Comparison of selected rainfall events

**Lines 319-333:** "Hydrological calibration includes uncertainties due to conceptual simplification, processes not incorporated in the model, and unknown processes to the modeler

[revised manuscript text omitted]

**Comment #16**

Line 336 Figure 5: Again, I am not sure these simulations are actually based on which precipitation resolution and runoff method.

Reply: Figure 4 shows the observed and simulated hourly hydrographs of selected high rainfall events that occurred in the years 2018 and 2019 using the GAML method. We will emphasize the method in the figure caption.

Line 336: "Figure 4. Observed and simulated hourly discharge ($m^3$ $s^{-1}$) using the GAML method for the heavy rainfall events that occurred in 2018 and 2019: (a) event from 12/01-14/01/2018; (b) event from 05/05-07/05/2018; (c) event from 29/09-01/10/2018; (d) event from 05/02-07/02/2019; (e) event from 12/11-14/11/2019; (f) event from 29/12-31/12/2019."

---

## Author Response (AR3)

National Technical University of Athens
School of Mining & Metallurgical Engineering
Laboratory of Engineering Geology & Hydrogeology

Dear Editor,

We would like to submit the revised version of our paper "Hydrological modeling using the SWAT Model in urban and peri-urban environments: The case of Kifissos experimental sub-basin (Athens, Greece)" by Evgenia Koltsida, Nikos Mamassis, and Andreas Kallioras.

We would also like to thank the referees for their time spent and effort on reviewing our manuscript. We appreciate their valuable comments and suggestions, which helped us in improving the quality of the manuscript.

We confirm that this manuscript has not been published elsewhere and is not under consideration by another journal. The authors have no conflicts of interest to disclose and have all approved this submission.

Thank you for your time and consideration.

Sincerely,
Evgenia Koltsida, M.Sc.